# Co-movement of astral microtubules, organelles and F-actin by dynein and actomyosin forces in frog egg cytoplasm

James F Pelletier[1,2,3], Christine M Field[1,2], Sebastian Fürthauer[4], Matthew Sonnett[1], Timothy J Mitchison[1,2]*

[1]Department of Systems Biology, Harvard Medical School, Boston, United States; [2]Marine Biological Laboratory, Woods Hole, United States; [3]Department of Physics, Massachusetts Institute of Technology, Cambridge, United States; [4]Flatiron Institute, Center for Computational Biology, New York, United States

**Abstract** How bulk cytoplasm generates forces to separate post-anaphase microtubule (MT) asters in *Xenopus laevis* and other large eggs remains unclear. Previous models proposed that dynein-based, inward organelle transport generates length-dependent pulling forces that move centrosomes and MTs outwards, while other components of cytoplasm are static. We imaged aster movement by dynein and actomyosin forces in *Xenopus* egg extracts and observed outward co-movement of MTs, endoplasmic reticulum (ER), mitochondria, acidic organelles, F-actin, keratin, and soluble fluorescein. Organelles exhibited a burst of dynein-dependent inward movement at the growing aster periphery, then mostly halted inside the aster, while dynein-coated beads moved to the aster center at a constant rate, suggesting organelle movement is limited by brake proteins or other sources of drag. These observations call for new models in which all components of the cytoplasm comprise a mechanically integrated aster gel that moves collectively in response to dynein and actomyosin forces.

**\*For correspondence:**
Timothy_Mitchison@hms.harvard.edu

**Competing interests:** The authors declare that no competing interests exist.

## Introduction

Cytokinesis requires drastic reorganization of the cell and provides a window into cytoplasmic mechanics and principles of sub-cellular organization. Here, we focus on organization of the cytoplasm by MT asters between mitosis and cytokinesis in *Xenopus laevis* eggs. The large size of eggs and the availability of an optically tractable egg extract system make *Xenopus* a good model for analysis of cytoplasmic organization. The first mitotic spindle is centrally located and much smaller than the egg. After mitosis, a pair of MT asters grow out from the centrosomes, reaching the cortex ~20 min later. These asters are composed of a branched network of short, dynamic MTs ~15 µm long and oriented approximately radially, with plus ends outward (*Ishihara et al., 2016*; *Ishihara et al., 2014*). Interphase egg asters have several organizational and mechanical functions. Where the paired asters meet, at the midplane of the egg, the MTs form an antiparallel interaction zone which recruits the chromosomal passenger complex (CPC) and centralspindlin (*Field et al., 2015*). In this way, a pair of asters defines the cleavage plane (*Basant and Glotzer, 2018*; *Carmena et al., 2012*). The focus of this paper is on how asters move centrosomes and nuclei away from the future cleavage plane, so each daughter blastomere inherits one of each. This separation movement transports centrosomes and nuclei hundreds of microns away from the midplane over tens of minutes, as illustrated in *Figure 1A–C*. In common with other authors, we often refer to centrosome and aster movement as the same process. The reality is more complex due to continuous MT growth and turnover. As centrosomes move away from the midplane, the interaction zone

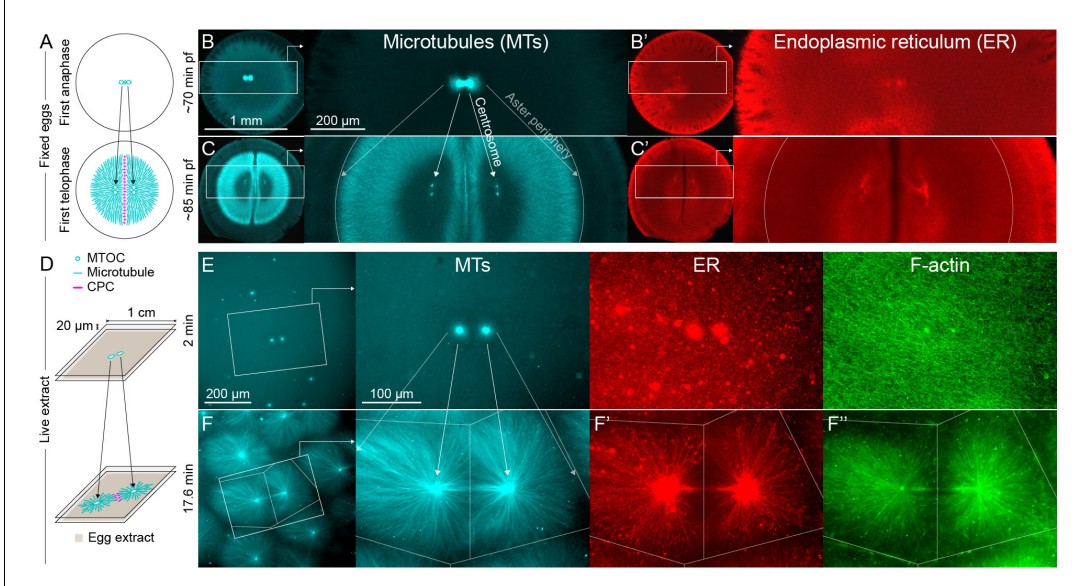

**Figure 1.** MTOC separation movement in *Xenopus* eggs and egg extract. Panels A-C are fixed embryos, and panels D-F are in *Xenopus* egg extract. (A) Cartoon illustrating MTOC movement away from the CPC-positive midplane before astral microtubules (MTs) reach the cortex in *Xenopus laevis* eggs. MTs shown in cyan and CPC-positive interaction zone in magenta. Note the CPC is shown in the cartoon panels A and D, but not in the rest of the figure. (B,C) Anti-tubulin immunofluorescence of eggs fixed ~70 and ~85 min post-fertilization (pf). Diagonal lines connecting different eggs in panels B and C emphasize centrosome separation movement and the growing aster periphery. (B',C') Anti-LNPK (ER) immunofluorescence of the same eggs. (D) Cartoon illustrating aster separation movement in an extract system. MTs and CPC as in panel A. Asters were reconstituted from artificial microtubule organizing centers (MTOCs) in interphase *Xenopus* egg extracts. (E,F) MTOCs moved apart as asters grew and interacted with one another over time. Time is defined with respect to perfusing the sample and warming to 20°C, so the start of aster growth occurred soon after 0 min. (F') A fraction of the ER became enriched around MTOCs, and (F'') F-actin was disassembled locally along interaction zones.

The online version of this article includes the following figure supplement(s) for figure 1:

**Figure supplement 1.** Higher magnification imaging around MTOCs included signatures of both co-movement and relative movement of astral MTs, ER, and F-actin.

between the asters remains stationary, while the outer aster periphery grows outwards due to a combination of MT polymerization and outward sliding.

The forces that act on MTs to move asters, centrosomes, and nuclei differ between systems (*Garzon-Coral et al., 2016*; *Grill and Hyman, 2005*; *Kotak and Gönczy, 2013*; *Meaders and Burgess, 2020*; *Xie and Minc, 2020*). Centration movement of the sperm aster after fertilization and movement of sister asters away from the midzone after mitosis are thought to involve similar mechanics (ibid). Our focus is on post-mitotic movement to avoid the complication of MT-plasma membrane interactions. In *Xenopus* and zebrafish zygotes, which are unusually large cells, aster movement away from the midplane is driven by dynein-dependent pulling forces (*Wühr et al., 2010*). Since movement occurs before astral MTs reach the cortex, the dynein must be localized throughout the cytoplasm, presumably attached to organelles, but this was not tested. The most prominent model for aster movement of this kind proposes that dynein attached to organelles throughout the aster generates pulling forces that increase with MT length (*Hamaguchi and Hiramoto, 1986*; *Kimura and Kimura, 2011*; *Tanimoto et al., 2016*; *Tanimoto et al., 2018*; *Wühr et al., 2010*). In this 'length-dependent pulling' model, dynein transports organelles along astral MTs toward the centrosome, then viscous or elastic drag on the organelles imparts a counter force on the MTs, pulling them away from the centrosome. The flux of organelles, and thus the net pulling force, is thought to scale with MT length. Although length-dependent pulling models are widely discussed, many aspects remain unclear, for example, net forces may not scale with MT length due to hydrodynamic interactions between MTs (*Nazockdast et al., 2017*). The organelles that anchor dynein in the cytoplasm of large egg cells have not been fully identified and the spatiotemporal distribution of organelle transport has not been mapped. Candidate dynein anchor organelles include the ER, which moves inwards as sperm asters center in sea urchin (*Terasaki and Jaffe, 1991*), acidic organelles which were

implicated in nematode eggs (*Kimura and Kimura, 2011*) and mitochondria which are abundant in early embryos.

Contractile activity of actomyosin can cause centrosome and aster movement in eggs and embryos (*Field and Lénárt, 2011*; *Telley et al., 2012*), but its contribution to centrosome separation movement in *Xenopus* eggs is unclear. Bulk cytoplasmic F-actin is a major mechanical element in *Xenopus* eggs (*Elinson, 1983*) and egg extracts (*Field et al., 2011*). In egg extracts, F-actin can impede centrosome movement in meiotic extracts (*Colin et al., 2018*), but F-actin is not required for centrosome separation movement in cycling extracts (*Cheng and Ferrell, 2019*). Caution is required when extrapolating from drug studies to the mechanics of unperturbed cytoplasm. F-actin depolymerization softens the cytoplasm and presumably decreases the drag on moving asters as well as dynein anchors. Thus, F-actin depolymerization may modulate dynein-based forces on asters, in addition to removing actomyosin-based forces. Furthermore, effects of cytochalasins in *Xenopus* eggs are hard to interpret because they only permeate the *Xenopus* egg cortex during first cleavage, when new membrane becomes exposed (*de Laat et al., 1973*).

Considering both the length-dependent pulling model, and the role of actomyosin, one important question has not been rigorously addressed in any system: do centrosomes and associated astral MTs move *through* a static cytoplasm, as predicted by the length-dependent pulling model? Or do they move *with* other components of cytoplasm, such as organelles, F-actin and cytosol? If organelles anchor dynein, the length-dependent pulling model predicts that they must move in the opposite direction as the centrosome, or at least remain stationary. Inward movement of organelles in moving asters has been reported in some systems (*Hamaguchi and Hiramoto, 1980*; *Hamaguchi and Hiramoto, 1986*; *Kimura and Kimura, 2011*; *Terasaki and Jaffe, 1991*), but to our knowledge, there are no quantitative studies relating organelle flux to forces on asters. The dynamics of F-actin and cytosol in moving asters have not been addressed to our knowledge. Theoretical models of the length-dependent force model implicitly assume that these components are static and homogenous and contribute to viscous drag on moving asters (*Tanimoto et al., 2016*; *Tanimoto et al., 2018*). Live observation of multiple components of cytoplasm is required to address these questions. This is not possible in opaque frog eggs so we turned to actin-intact egg extracts. Growth and interaction of interphase asters were previously reported in this system (*Ishihara et al., 2014*; *Nguyen et al., 2014*), but aster movement was not systematically investigated. Here, we report methods for observing aster movement in egg extract and use them to measure relative movement of MTs, organelles, and F-actin. We observed all cytoplasmic networks mostly moved together inside asters. The highest velocity differences between networks occurred at the aster periphery. Mechanical integration between all cytoplasmic components inside asters requires new models for aster movement.

## Results

### Centrosome separation and ER distribution in fixed eggs

As a first test of how centrosomes and organelles move relative to one another, we fixed frog eggs before first cleavage, stained for tubulin and ER, and imaged by confocal microscopy (*Figure 1A–C*). Centrosome separation movement is represented by the diagonal lines connecting different eggs in panels B and C. As centrosomes move away from the midplane, the centrioles within them replicate and split, visible as the pair of bright cyan spots within each aster in *Figure 1C*. We probed the ER distribution by staining for the ER membrane marker Lunapark (LNPK) (*Figure 1B',C'*). The ER luminal marker PDIA3 had a similar distribution (not shown). The ER was distributed all over the asters, with some enrichment near centrosomes and the cortex. Lack of strong ER enrichment at centrosomes called into question the length-dependent pulling model with the ER as a dynein anchor. However, organelle transport dynamics could not be measured from fixed images, so we turned to an egg extract system for live imaging.

### Microtubule organizing center (MTOC) separation movement in egg extract by dynein and actomyosin

To model aster separation movement in a cell-free system suitable for live imaging, we filled chambers consisting of two PEG-passivated coverslips spaced ~20 μm apart with actin-intact interphase

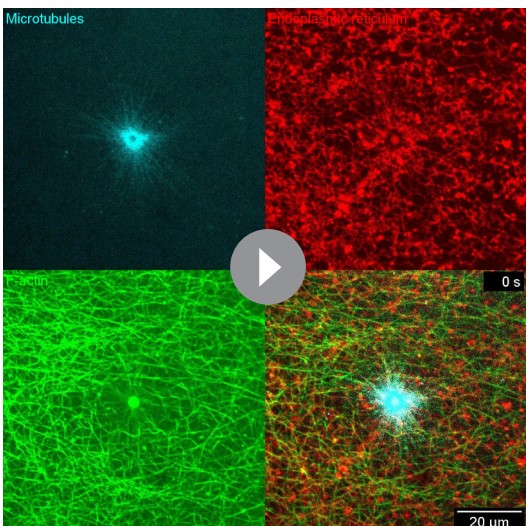

**Video 1.** Dynamic reorganization of cytoplasmic networks during the initial stages of aster nucleation and growth imaged at 60x. (Related to *Figure 1—figure supplement 1*) MTs were labeled with tubulin-Alexa Fluor 647, ER with DiI, and F-actin with Lifeact-GFP. Imaged on a spinning disk confocal with 60x objective lens. Cytoplasmic networks were highly dynamic, and astral MTs dynamically reorganized the ER and F-actin networks. Parts of the ER exhibited abrupt and transient motion toward the MTOC, presumably driven by dynein, and the F-actin transitioned from random to radial entrainment with MTs.

https://elifesciences.org/articles/60047#video1

(*Figure 2A*) as previously reported (*Nguyen et al., 2014*), modeling similar zones in eggs (*Field et al., 2015*). CPC-positive interaction zones cause local disassembly of both MTs and F-actin, which locally softens the cytoplasm (*Field et al., 2019*). The resulting anisotropies in MT and F-actin density may lead to generation of directed forces on MTOCs by both dynein and actomyosin.

To quantify MTOC movement, and determine the role of forces from different motors, we picked random locations and imaged large fields over time in up to four conditions in parallel. *Figure 2B* and *Video 3* show a typical experiment, where only the CPC channel is shown for simplicity. At early times points, the spatial distribution of MTOCs was random and the CPC signal was diffuse, except some signal on the MTOCs. As asters grew and interacted, they recruited CPC to zones between them under all conditions. We quantified MTOC movements with respect to their nearest neighbors, which were defined by the Delaunay triangulation

egg extract containing artificial MTOCs, imaging probes and drugs. We then imaged aster growth and movement over ~30 min. For most experiments we used widefield microscopy with a 20x objective to collect data on overall organization and flows, in some cases stitching multiple image fields. To illustrate structural details of the components we studied, *Figure 1—figure supplement 1* and *Video 1* show MTs, ER, and F-actin near an MTOC by spinning disk confocal microscopy with a 60x objective. In 20x magnification fields, we routinely noted that MTOCs that were close together at early time points tended to move apart. *Figure 1D* illustrates the extract system, and *Figure 1E,F* show an example of MTOCs moving apart as asters grew and interacted with one another. This kind of separation movement was observed in hundreds of image sequences, such as in *Video 2*, and we believe it models centrosome separation movement in eggs.

When asters grew to touch each other, they formed CPC-positive interaction zones

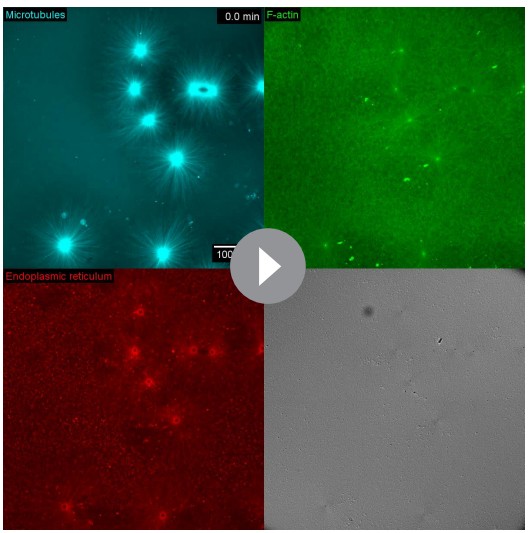

**Video 2.** Co-movement of MTs, ER, and F-actin during aster separation movement. (Related to *Figures 2A* and *3*) MTs were labeled with tubulin-Alexa Fluor 647, ER with DiI, F-actin with Lifeact-GFP, and organelles were shown in differential interference contrast (DIC). All cytoplasmic networks moved together. Note the flow of organelles visible in DIC: inside asters, where the density of F-actin, MTs, and ER was higher, organelles flowed with the asters; whereas along interaction zones between asters where the density of F-actin was lower, organelles flowed in the opposite direction, into the space on the right that was vacated by the asters moving to the left.

https://elifesciences.org/articles/60047#video2

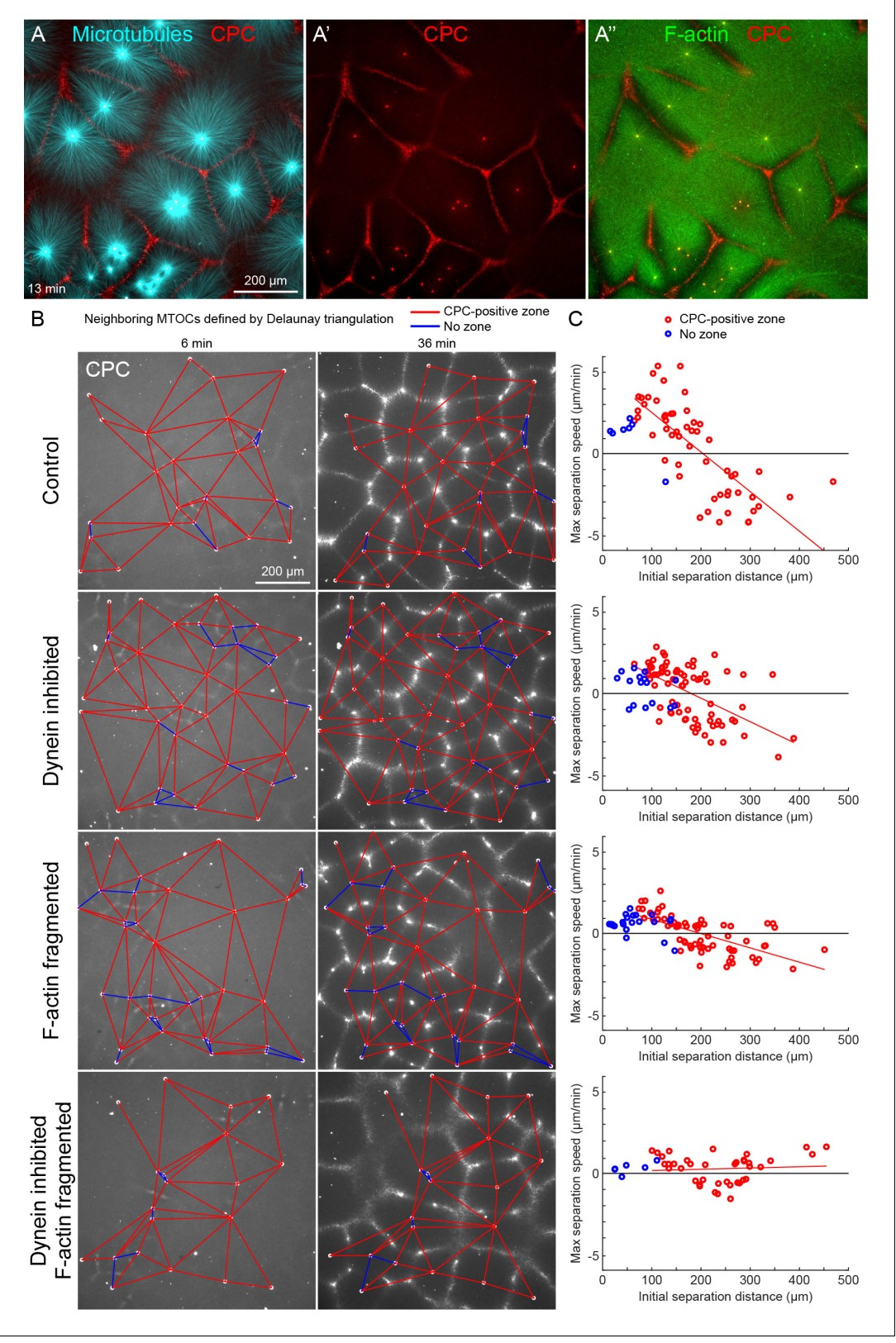

**Figure 2.** MTOC separation movement in egg extract by dynein and actomyosin. (**A**) The CPC localized to interaction zones between neighboring asters, blocking mutual interpenetration of MTs and disassembling F-actin locally. Time is defined with respect to perfusing the sample and warming to 20°C, so the start of aster growth

*Figure 2 continued on next page*

*Figure 2 continued*
occurred soon after 0 min. (B) Four aster growth reactions were followed in parallel under control vs inhibitor conditions. The first column in each condition shows an early time point, and the second column shows a time point 30 min later. MT growth was similar and CPC-positive interaction zones formed under all conditions (see *Video 3*). (C) Maximum speed of separation with respect to initial distance between the MTOCs. Red lines indicate linear fits to red points.

The online version of this article includes the following source data for figure 2:

**Source data 1.** MTOC trajectories and Delaunay triangulations for panels B and C.

between MTOCs at the earliest time point and followed over the video (*Figure 2B*). Red edges indicate when neighboring MTOCs formed a CPC-positive interaction zone between them, and blue edges indicate when they did not. We then measured the maximum separation speed as a function of the initial separation distance between the MTOCs. Under control conditions, MTOCs that were initially closer together tended to move farther apart, while those initially farther apart tended to move closer together, leading to MTOCs becoming more regularly spaced at the end of the sequence. This directionality is evident from the strong negative correlation between the maximum speed of separation movement and starting distance (*Figure 2C*). We focused on separation movement of MTOCs in separate asters with a CPC-positive interaction zone between them (red points), since this models post-anaphase centrosome separation movement in eggs.

To test the role of dynein and actomyosin in MTOC movement, we inhibited dynein using the p150-CC1 fragment of dynactin (*King et al., 2003*) or fragmented F-actin using Cytochalasin D. Inhibiting either motile system alone caused a partial block to aster movement, and inhibiting both caused an almost complete block (*Figure 2C*). Inhibiting CPC recruitment with an AURKB inhibitor also completely blocked MTOC movement (not shown). The contributions of dynein and actomyosin forces to aster movement were similar, as judged by similar effects of single inhibition on the slopes of separation speed vs initial distance plots (*Figure 2C*). These findings were qualitatively confirmed by visual inspection and partial analysis of more than 10 experiments using multiple extracts. We interpret these data as showing that MTOC movement in our extract system is driven by a combination of dynein and actomyosin forces. We investigate sites of dynein-based pulling below. We hypothesize actomyosin-based separation movement is driven by actomyosin contraction away from regions of lower F-actin density along interaction zones and will analyze this model in detail elsewhere. With a reliable system for aster separation movement in hand, we next interrogated organelle and F-actin dynamics.

**Video 3.** Both dynein and actomyosin contribute to aster separation movement. (Related to *Figure 2*) We compared four conditions: control with F-actin intact, dynein inhibited by p150-CC1, F-actin fragmented by Cytochalasin D, and double inhibition of dynein and F-actin. F-actin was labeled with Lifeact-GFP, ER was labeled with DiI, organelles were shown in differential interference contrast (DIC), and CPC-positive interaction zones were labeled with anti-INCENP-Alexa Fluor 647. MTs grew and CPC-positive interaction zones formed between asters in all conditions. F-actin and ER were imaged instead of MTs because local disassembly of F-actin along CPC-positive interaction zones is thought to help aster separation movement, and inward transport of ER and other organelles is thought to drive dynein-based aster movement.

https://elifesciences.org/articles/60047#video3

## ER and F-actin move with MTs in separating asters

Aster separation trajectories were longest, and most unidirectional, when MTOCs were clustered at the initial time point. In these cases, MTOCs moved predictably outwards from the cluster as

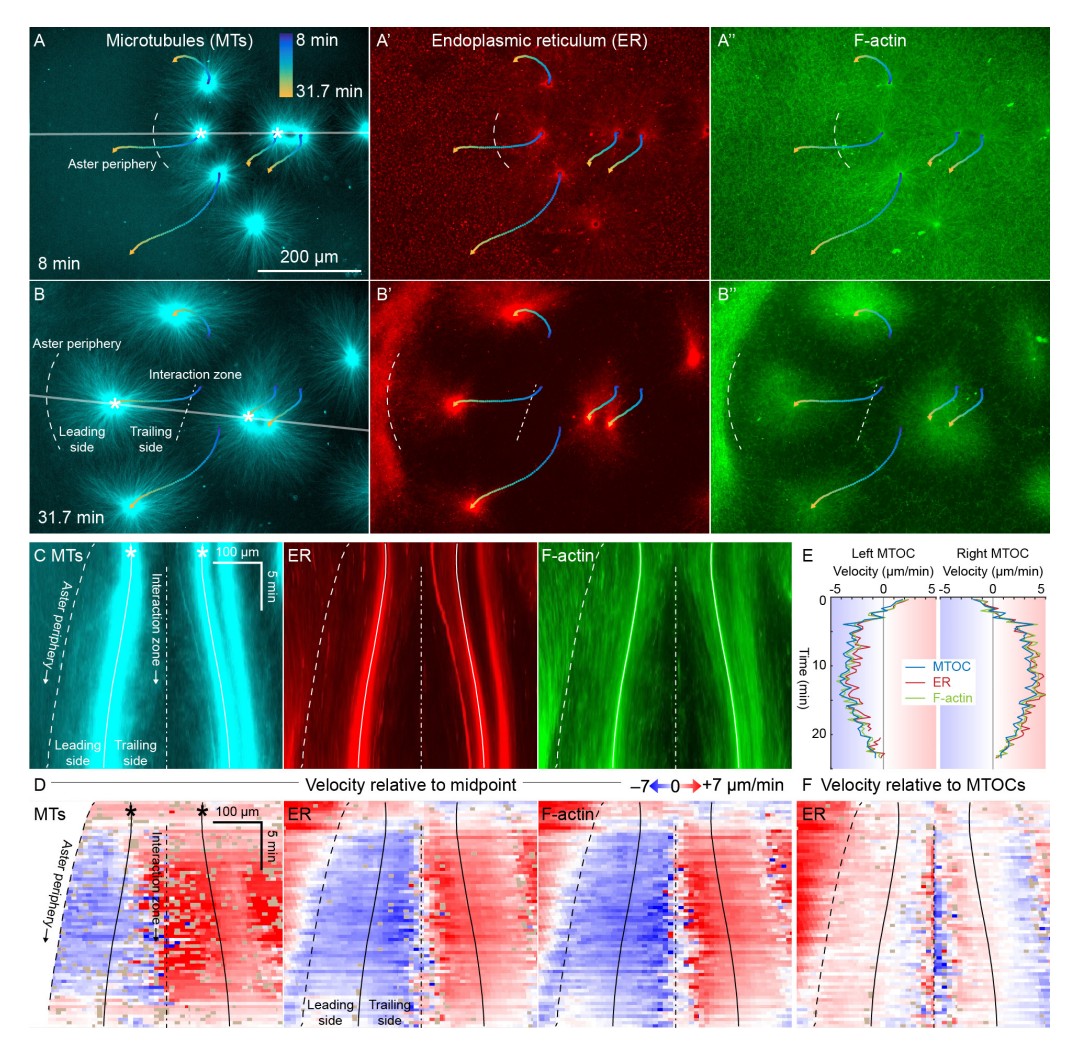

**Figure 3.** ER and F-actin move with MTs in separating asters. (**A,B**) Asters grew until they reached their neighbors, formed interaction zones approximately equidistant between the MTOCs, then moved away from the interaction zones (see *Video 2*). MTOC trajectories are represented by contours colored from blue to yellow. Time is defined with respect to perfusing the sample and warming to 20°C, so the start of aster growth occurred soon after 0 min. (**C**) Intensity kymographs along the gray line shown in panels A and B, passing through the MTOCs marked with a white star. To show relative movement of the MTOCs, each row of the kymograph was computationally translated to keep stationary the midpoint between the MTOCs, where the interaction zone formed. Solid curves indicate the MTOCs, the dashed curve indicates the growing aster periphery, and the dash-dotted line indicates the interaction zone. (**D**) Velocity maps in the same frame of reference as in panel C. 2D flow fields were measured by particle image velocimetry (PIV), projected onto the line passing through the MTOCs, then the projected velocity of the midpoint between the MTOCs was subtracted, again to show movement relative to the interaction zone. A white color indicates stationary with respect to the midpoint, blue indicates moving to the left, and red to the right. PIV outliers were filtered and shown in beige. (**E**) Velocity of the MTOCs based on particle tracking, as well as the velocity of ER and F-actin in the neighborhood of the MTOCs based on PIV. (**F**) Velocity of ER with respect to the moving MTOCs, not with respect to the interaction zone as in panel D.

The online version of this article includes the following source data and figure supplement(s) for figure 3:

**Source data 1.** Velocity maps for panel D, as well as the full 2D velocity fields from PIV used to generate the velocity maps.

**Figure supplement 1.** Higher magnification imaging around zones included signatures of both co-movement and relative movement of astral MTs, ER, and F-actin.

asters grew out and CPC-positive interaction zones formed between them (*Figure 3A,B*, *Video 2*). In *Figure 3A*, future MTOC trajectories are superimposed on an early time point to illustrate separation movement. To investigate how ER and F-actin moved with respect to moving astral MTs, we first used kymograph analysis. We picked a pair of MTOCs that moved apart, indicated by stars in

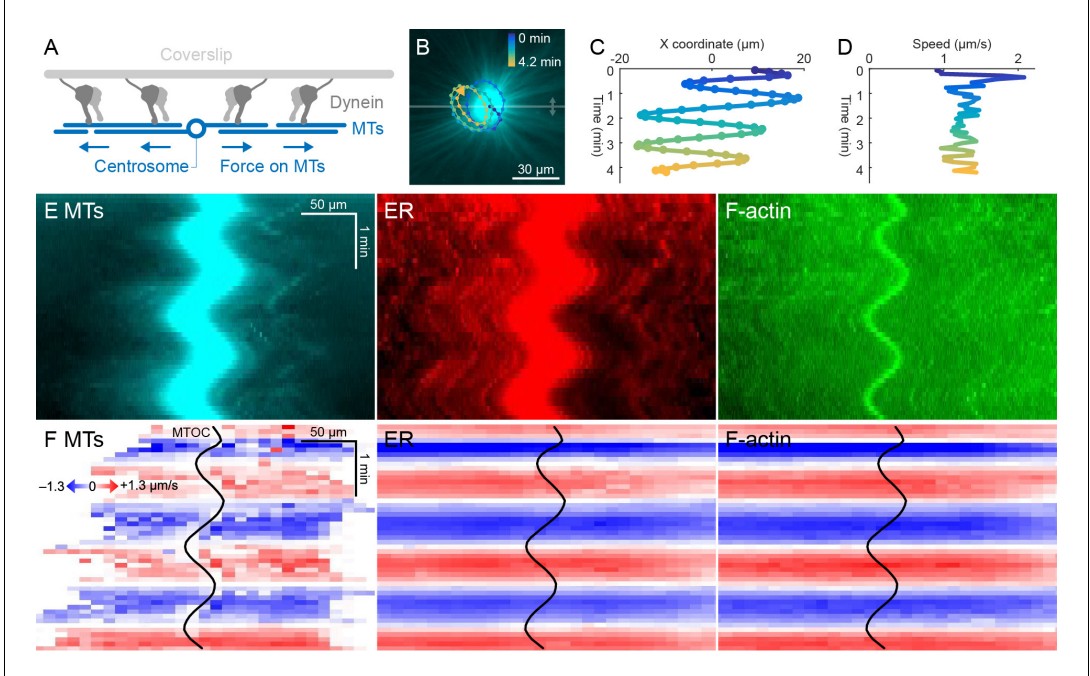

**Figure 4.** ER and F-actin move with MTs on coverslips functionalized with dynein. (**A**) Coverslips were functionalized with an antibody against HOOK2, so the rigid coverslip substrate generated pulling forces on the astral MTs. (**B**) Circular oscillatory trajectory of the MTOC (see *Video 5*). (**C**) X coordinate of the MTOC. (**D**) Speed of the MTOC relative to the coverslip, including both X and Y components of motion. (**E**) Intensity kymographs along the horizontal line passing through the MTOC, indicated in panel B. (**F**) Velocity maps in the same frame of reference as in panel E. 2D velocity fields were measured by particle image velocimetry (PIV) then projected onto the horizontal line as in panel E. The MTOC position is shown as a black curve.

The online version of this article includes the following source data and figure supplement(s) for figure 4:

**Source data 1.** Velocity maps for panel F, as well as the full 2D velocity fields from PIV used to generate the velocity maps.
**Figure supplement 1.** Characterization of the HOOK2 C-terminal peptide antibody.
**Figure supplement 1—source data 1.** Immunoprecipitation-mass spectrometry (IP-MS) counts for *Figure 4—figure supplement 1*.

---

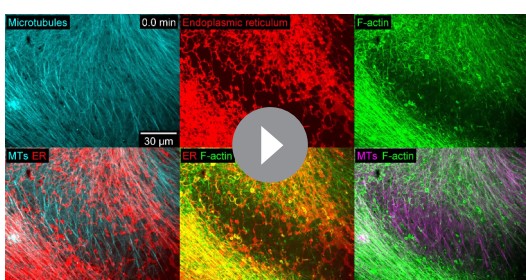

**Video 4.** Signatures of both co-movement and relative movement in moving asters imaged at 60x. (Related to *Figure 3—figure supplement 1*) MTs were labeled with tubulin-Alexa Fluor 647, ER with DiI, and F-actin with Lifeact-GFP. Imaged on a spinning disk confocal with 60x objective lens. All networks were highly dynamic. Some ER and F-actin features moved relative to astral MTs, deformed, or otherwise changed structure, which provide examples where co-movement breaks down on small spatiotemporal scales; however, the dominant trend was co-movement of the networks.
https://elifesciences.org/articles/60047#video4

*Figure 3A–C*. Then we generated kymographs in all channels (*Figure 3C*) along the line passing through the MTOCs, indicated by the grey line in *Figure 3A,B*. Visual inspection revealed features in all three channels that tracked parallel to the separating MTOCs, suggesting all the networks were moving together away from the interaction zone, on both the leading and trailing sides of the aster indicated in *Figure 3B*. Organelles visible in differential interference contrast (DIC) images also moved away from the interaction zone (*Video 2*). These features are most evident in the F-actin kymograph, but can be seen in all channels by magnifying the figure and inspecting closely. Visual inspection and kymograph analysis of image sequences from more than 10 independent experiments confirmed that all components of asters tend to move together during separation movement, and that the data in *Figure 3* are typical.

To better quantify movement of MTs, ER, and F-actin as asters separated, we measured 2D flow

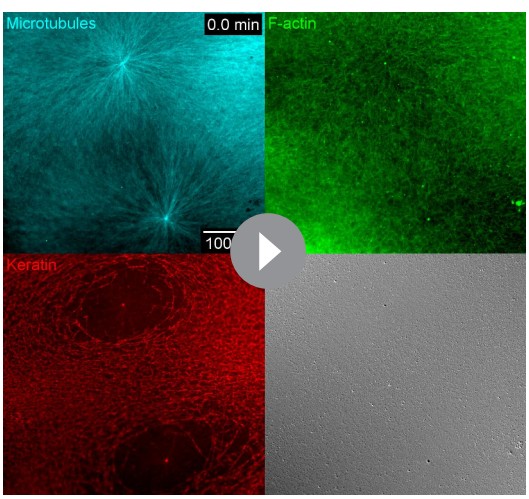

**Video 5.** Co-movement of MTs, ER, and F-actin during oscillatory aster movement on coverslips functionalized with dynein. (Related to *Figure 4*) MTs were labeled with tubulin-Alexa Fluor 647, ER with DiI, and F-actin with Lifeact-GFP. All cytoplasmic networks moved together. Dynein was recruited to coverslips via an antibody to the endogenous dynein adapter HOOK2.
https://elifesciences.org/articles/60047#video5

fields by particle image velocimetry (PIV). All three cytoplasmic networks moved in the same direction at similar speeds of up to 7 μm/min. Movement was always directed away from the interaction zone on both leading and trailing sides of the aster, as shown by blue on both sides annotated in *Figure 3C,D*. This is inconsistent with the length-dependent pulling model, in which organelles on the leading side must move toward the interaction zone (red) or at least remain stationary with respect to the interaction zone (white). *Figure 3E* compares the MTOC velocity from particle tracking to the ER and F-actin velocities from PIV, again consistent with all three cytoplasmic networks moving outwards at similar speeds. To highlight relative movement within asters, *Figure 3F* shows the ER velocity rel-ative to the MTOC velocity. MTOCs and ER moved outwards at similar rates near the center of the asters, as evidenced by the pale colors in *Figure 3F*. In contrast, there was more relative movement at the external and internal peripheries (*Figure 3D,F*). ER movement relative to MTs at the aster periphery is investigated in detail below.

Close inspection of *Figure 3D* and similar analyses showed that velocities of all three networks away from the interaction zone were not constant throughout the aster, though different networks had similar velocities at any given location. Typically, the region near the interaction zones moved ~20% faster than the MTOC, and the leading edge of each aster moved ~20% slower. This spatial variation in velocity shows that the aster does not move as a completely rigid body. Rather, it deforms as a gel, locally compressing or stretching in response to forces and stresses.

## Higher magnification imaging shows saltatory as well as correlated movement

Co-movement of cytoplasmic networks in image sequences collected with a 20x objective is in apparent disagreement with mechanisms known to transport networks relative to one another, for example, by motors or tip tracking (*Lane and Allan, 1999*; *Wang et al., 2013*; *Waterman-Storer et al., 2000*; *Waterman-Storer et al., 1995*). To resolve this discrepancy, we imaged asters at higher spatiotemporal resolution using 60x spinning disk confocal microscopy (*Figure 1—figure supplement 1* and *Figure 3—figure supplement 1*, *Videos 1* and *4*). At 60x, we observed that some local segments of the ER exhibited rapid, saltatory movement toward and away from the MTOC (*Figure 1—figure supplement 1B'*) as well as rapid, transient deformation of the ER (*Figure 3—figure supplement 1D*). Although these examples show that co-movement can break down on small spatiotemporal scales, we observed a predominance of co-movement even at 60x. In tangential kymographs, pivoting movement of MTs was mirrored by ER and F-actin features (*Figure 1—figure supplement 1C*). In radial kymographs, MT speckles, ER, and F-actin features slid outwards together, likely driven by dynein because dynein inhibition blocks such outward MT sliding (*Ishihara et al., 2014*).

**Video 6.** Co-movement of keratin with moving asters during oscillatory aster movement. (Related to *Figure 4*) MTs were labeled with tubulin-Alexa Fluor 647, F-actin with Lifeact-GFP, and keratin with anti-keratin-Alexa Fluor 568. All cytoplasmic networks moved together.
https://elifesciences.org/articles/60047#video6

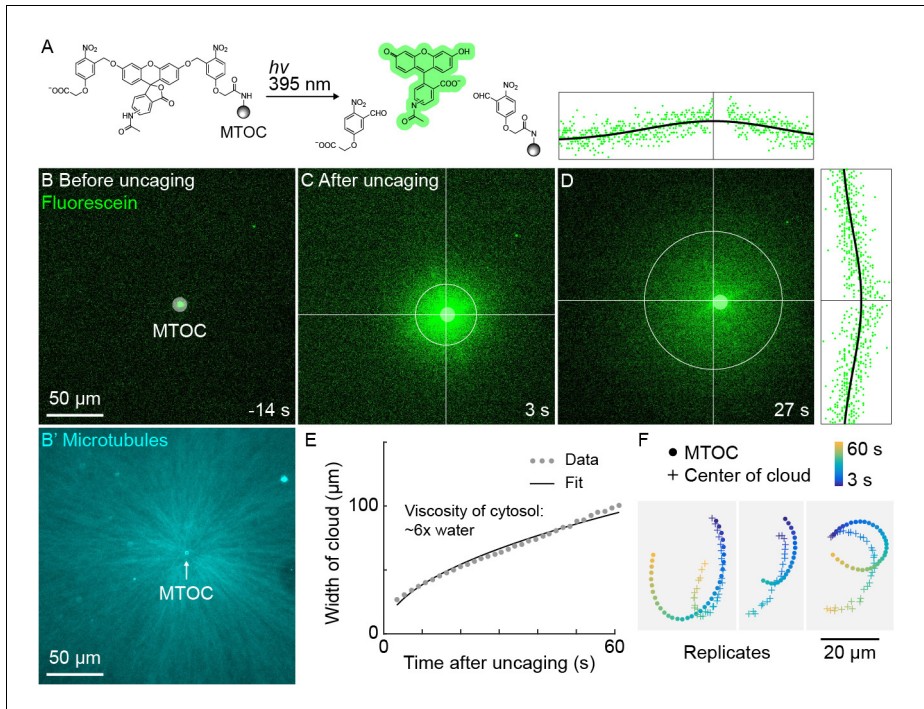

**Figure 5.** A small molecule is advected with moving asters. (**A**) To track the flow of a small molecule within moving asters, MTOCs were functionalized with caged fluorescein. (**B**) Caged fluorescein, before uncaging. (**B'**) Astral MTs radiating from the MTOC filled the region. The aster was oscillating on a coverslip functionalized with anti-HOOK2 as in *Figure 4*. (**C**) Fluorescein, after uncaging. (**D**) Within tens of seconds, the fluorescein diffused away from the MTOC and approached the background intensity (see *Video 7*). 2D Gaussian fits to estimate the width and center of the fluorescein cloud. The bright MTOC was excluded from the Gaussian fit, so uncaged fluorescein that remained bound to the MTOC did not bias the fitted position. (**E**) Expansion of the fluorescein cloud width fit to a model of diffusion. (**F**) Several replicate trajectories of the MTOC (circle) and the center of the fluorescein cloud (plus).

The online version of this article includes the following source data and figure supplement(s) for figure 5:

**Figure supplement 1.** Hypothetical constant flow permeating asters can improve registration between MTOC and center of fluorescein cloud.

**Figure supplement 1—source data 1.** Width of the fluorescein cloud vs time for panel E, and MTOC and cloud center trajectories for panel F.

**Figure supplement 2.** Separating asters exhibited saddle-shaped flow fields, consistent with advection of cytosol by moving asters.

In moving asters, intensity features in all three networks largely tracked together even at 60x (*Figure 3—figure supplement 1B*), confirmed by PIV velocity maps (*Figure 3—figure supplement 1C*). In summary, our data confirm literature reports that ER and F-actin can move rapidly relative to MTs on small spatiotemporal scales, but show that on scales of tens of microns and minutes, they tend to move together.

## ER and F-actin move with MTs on coverslips functionalized with dynein

To provide a complementary system for dynein-dependent MTOC movement, we artificially anchored dynein to the coverslip via a biologically relevant linkage. Endogenous HOOK2, a coiled-coil dynein-dynactin adapter (*Reck-Peterson et al., 2018*), was recruited to PEG-passivated coverslips via an antibody raised to its C-terminus (Materials and methods). To characterize the antibody and identify HOOK2 interacting proteins, we performed quantitative immunoprecipitation-mass spectrometry (IP-MS) (*Figure 4—figure supplement 1*). We compared three conditions: anti-HOOK2 in interphase extracts (three separate extract repeats), anti-HOOK2 in mitotic extracts (two repeats), and as negative control, random IgG in interphase extracts (three repeats). HOOK2 was

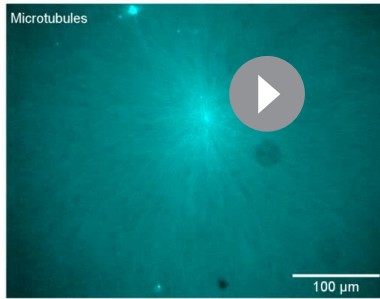

**Video 7.** Advection of fluorescein with moving asters during oscillatory aster movement. (Related to *Figure 5*) The first frames show MTs labeled with tubulin-Alexa Fluor 647, and the aster filled the region. The next few frames show the caged fluorescein attached to the MTOC. Then, the fluorescein was simultaneously photo-released from the MTOC as its fluorescence was uncaged, releasing a cloud of fluorescent fluorescein around the MTOC. The fluorescein cloud was fit with a 2D Gaussian. The center of the cloud is indicated at the intersection of the red and green lines, and the standard deviation of the cloud is indicated by the black circle. The plots above and to the right indicate the intensity values along the lines, and the black curves show the 2D Gaussian fit along the lines.
https://elifesciences.org/articles/60047#video7

the most abundant protein recovered on anti-HOOK2 beads. HOOK3 was also detected, consistent with heterodimerization between HOOK family members (*Redwine et al., 2017*; *Xu et al., 2008*). In interphase extracts, anti-HOOK2 pulled down multiple subunits of the dynein-dynactin complex, plus known interactors LIS1 and CLIP1. All these dynein-related proteins were greatly reduced in pulldowns from mitotic extracts, suggesting the interaction between HOOK2 and dynein-dynactin is negatively regulated by CDK1 activity. We concluded that the HOOK2 antibody offers a physiological linkage to dynein, and we proceeded to test its ability to serve as a dynein anchor for aster movement.

Dynein attached to coverslips via HOOK2 generated pulling forces on MTs directed away from the MTOC (*Figure 4A*). We previously reported that dynein non-specifically adsorbed to non-passivated coverslips increases the rate of aster growth due to outward microtubule sliding, but did not move MTOCs (*Ishihara et al., 2014*). Remarkably, on HOOK2-functionalized coverslips, asters exhibited rapid translational movement in a circular pattern with a diameter of 20–30 µm (*Figure 4B,C*, *Video 5*). During this movement, MTOCs moved continuously at ~1 µm/s, approximately 10-fold faster than the separation movements described above and comparable to the maximum speed of dynein (*Reck-Peterson et al., 2018*; *Figure 4D*). This 2D-oscillatory movement was observed in >10 different experiments using different batches of extract, and was blocked by dynein inhibition with p150-CC1. We plan to investigate the instability that causes circular motion elsewhere. Here, we used the rapid aster movement as an alternative system to study how ER and F-actin move with respect to moving MTs. *Figure 4E* shows intensity kymographs along a horizontal line that tracks up and down with the MTOC, analogous to the kymographs in *Figure 3C*. *Figure 4F* shows velocity maps analogous to those in *Figure 3D*. The intensity kymographs reveal many features that tracked with the MTOC, and the velocity plots show that indeed, all the cytoskeletal networks moved in the same direction, at the same speed, at any location inside the aster. In another experiment, keratin was also advected with moving asters (*Video 6*). From these observations, we conclude that cytoplasmic networks are mechanically integrated inside asters, and cytoplasmic networks move together with moving asters.

## A small molecule probe is advected with moving asters

The high speed and predictability of oscillatory aster movement on HOOK2-functionalized coverslips enabled us to ask whether the cytosol was advected with the moving cytoplasmic networks. This question was inspired by recent experiments showing that moving actomyosin gels advect cytosol in *Drosophila* embryos (*Deneke et al., 2019*). We functionalized artificial MTOCs with caged fluorescein, linked to the MTOCs via the caging group (*Figure 5A*). The fluorescein was uncaged upon shining 395 nm light, simultaneously activating its fluorescence and releasing it from the MTOCs (*Figure 5A–C*, *Video 7*). The cloud of photo-released fluorescein dispersed within tens of seconds (*Figure 5D*). Rapid diffusive spread of the cloud validated that the fluorescein behaves as a freely diffusing small molecule (*Figure 5E*) and enabled estimation of the viscosity of the cytosol at ~6 x that of water (Materials and methods), consistent with previous estimates (*Luby-Phelps, 1999*; *Valentine et al., 2005*). We then fit the fluorescein cloud with a 2D Gaussian to track its center of

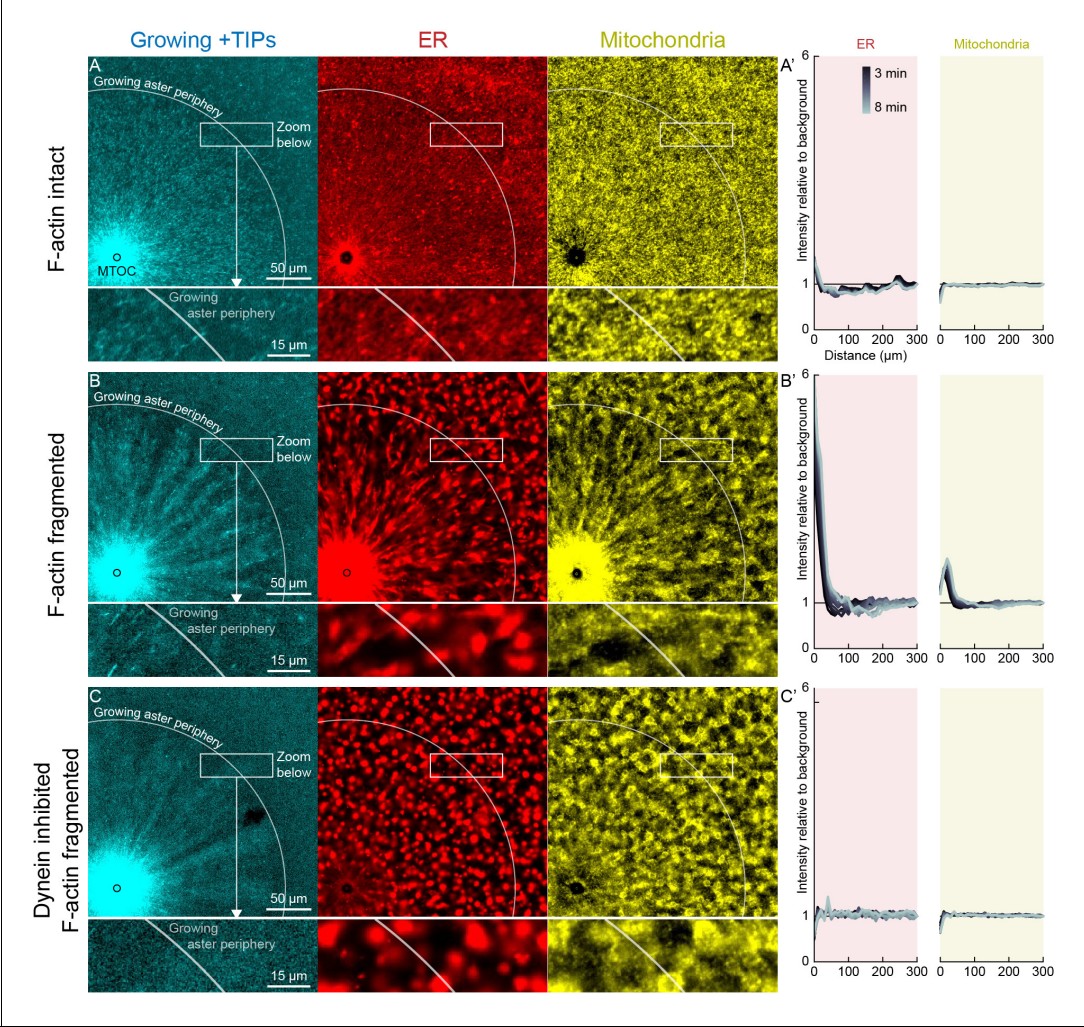

**Figure 6.** Dynein-mediated organelle movement is restricted by F-actin. (**A**) In control with intact F-actin, a small amount of ER became concentrated around the MTOC, but the majority of the ER and mitochondria remained distributed over the aster (see *Video 8*). The white arc indicates the growing aster periphery, and the box indicates the zoomed region in the lower panels. (**A'**) Average intensity with respect to distance from the MTOC over time, from black to gray. (**B**) When F-actin was fragmented with Cytochalasin D, a greater fraction of the ER was transported toward the MTOC, and a fraction of mitochondria was transported as well. Higher magnification: ER started to move when MTs indicated by growing +TIPs first grew into the cytoplasm, and ER and mitochondria co-localized with one another. (**C**) When dynein was inhibited with p150-CC1, the ER was not transported, neither toward nor away from the MTOC.

The online version of this article includes the following source data for figure 6:

**Source data 1.** ER and mitochondria intensity profiles for panels A', B', and C'.

---

mass, masking the bead so as not to bias the fit. The center of brightness of the diffusing fluorescein cloud and the MTOC had similar trajectories (*Figure 5F*), showing that cytosol advects with moving asters due to hydrodynamic interactions inside asters. Similar results were obtained in >10 experiments in three extracts. The cloud center did not precisely track with the moving MTOC, rather it tended to drift. Statistical analysis suggested this drift was probably not caused by tracking error (Materials and methods). We suspect that forces outside the aster can drive bulk flow of sol through the aster gel, carrying the diffusing fluorescein cloud with it. Consistent with this hypothesis, computationally translating the cloud center to remove the effect of a hypothetical constant flow field greatly improved registration between the cloud center and MTOC (*Figure 5—figure supplement 1*).

If separating asters on passivated coverslips also advect cytosol we would expect them to generate hydrodynamic forces and compensatory flows outside the asters. Disassembly of F-actin by CPC

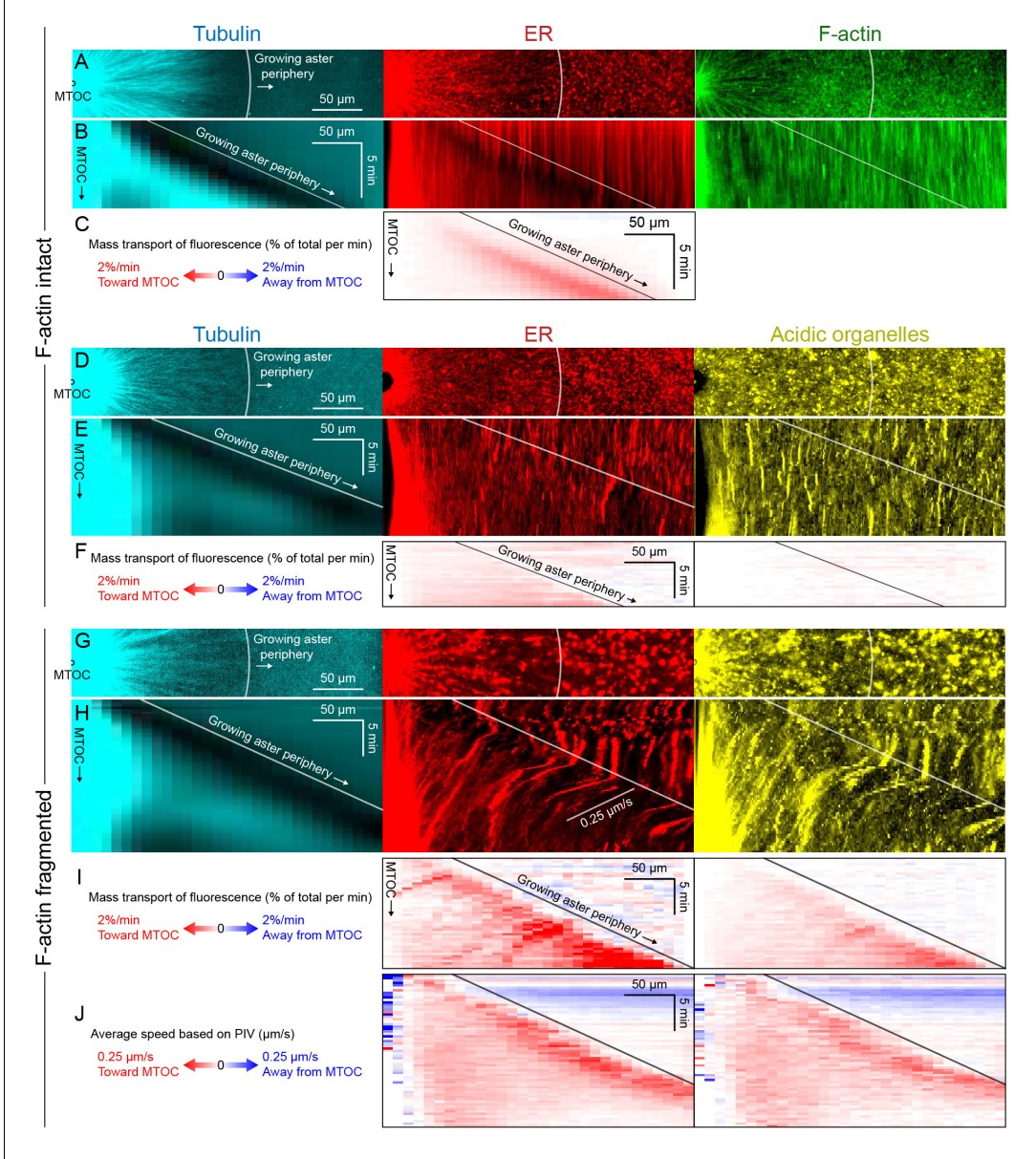

**Figure 7.** Dynein-mediated organelle movement is maximal on the aster periphery. (**A**) Stationary asters were grown from isolated MTOCs. The growing aster periphery is indicated by a white arc, and the ER was largely distributed but slightly depleted just inside the growing aster periphery. The ER exhibited a change in texture from slightly coarser outside the aster to finer inside the aster (see *Video 9*). (**B**) Kymographs along a line extending away from the MTOC. The MTOC corresponds to the left column, and the growing aster periphery corresponds to the diagonal line where soluble tubulin is depleted upon incorporation into the growing aster. (**C**) Mass transport map for ER averaged over a quadrant, in the same frame of reference as the kymographs in panel B. Mass transport analysis is described in *Figure 7—figure supplement 2*. (**D–F**) Similar experiment with F-actin intact, in a different batch of extract that exhibited less organelle movement. (**G–J**) Similar experiment with F-actin fragmented by Cytochalasin D (see *Video 10*). (**J**) Average speed based on PIV, in the same frame of reference as panels H,I and averaged over a quadrant. PIV is not shown for control because movement was too slow to be reliably quantified.

The online version of this article includes the following source data and figure supplement(s) for figure 7:

**Source data 1.** Mass transport maps for panels C, F, and I (in units of % of total per min), and PIV maps for panel J (in μm/s).

**Figure supplement 1.** Like other organelles, mitochondria exhibited a burst of organelle movement near the growing aster periphery.

**Figure supplement 2.** Explanation of flux analysis of organelle transport.

**Figure supplement 3.** Dextran was excluded in organelle-rich region within ~50 μm of MTOCs.

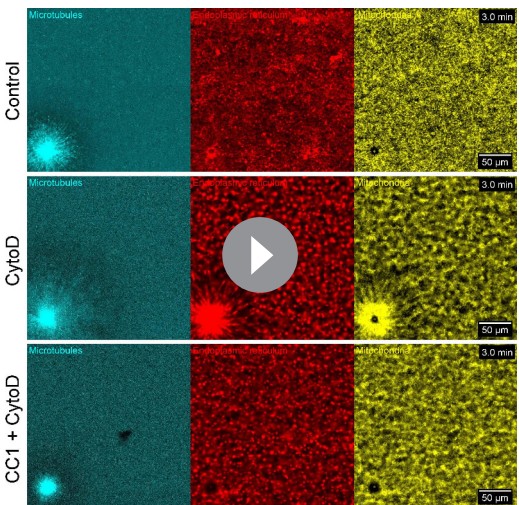

**Video 8.** F-actin reduced dynein-based transport of ER and mitochondria on stationary asters. (Related to *Figure 6*) The growing aster is indicated by growing +TIPs labeled with EB1-GFP, ER was labeled with DiI, and mitochondria with TMRE. In control with intact F-actin, some ER accumulated around the MTOC, and little to no mitochondria accumulated around the MTOC. When F-actin was fragmented, a greater fraction of ER and mitochondria were transported toward the MTOC. When dynein was inhibited, organelles were not transported, neither toward nor away from the MTOC.
https://elifesciences.org/articles/60047#video8

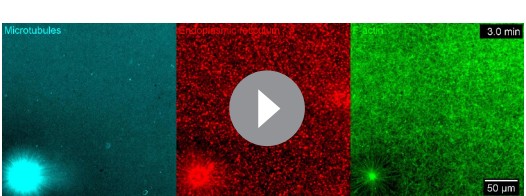

**Video 9.** Burst of ER movement at the growing aster periphery in control with F-actin intact. (Related to *Figure 7*) MTs were labeled with tubulin-Alexa Fluor 647, ER with DiI, and F-actin with Lifeact-GFP. The ER exhibited a burst of movement toward the MTOC at the growing aster periphery, resulting in transient depletion of the ER intensity near the aster periphery.
https://elifesciences.org/articles/60047#video9

makes the cytoplasm between separating asters more permeable to bulk flow of cytosol than the rest of the aster (*Field et al., 2019*). Thus, compensatory flows are expected to be directed inwards along interaction zones. We measured 2D flow fields around separating asters using PIV analysis of DIC image sequences. We indeed observed inward flow along the interaction zone (*Figure 5—figure supplement 2*). Advection of cytosol suggests moving asters constitute a poroelastic regime and places an upper bound of ~100 nm on their effective pore size (Materials and methods) (*Mitchison et al., 2008*; *Moeendarbary et al., 2013*).

## Dynein-mediated organelle movement is restricted by F-actin and interior MTs

Returning to passivated surfaces, we next investigated which organelles recruit dynein, and where they might exert forces that drive aster movement. To facilitate detailed analysis of organelle transport, we imaged isolated asters that remained stationary as they grew. ER and mitochondria are the most abundant organelles in *Xenopus* egg extracts based on proteomics (*Wühr et al., 2014*), and acidic organelles were implicated in centrosome movement in *C. elegans* embryos (*Kimura and Kimura, 2011*).

In control extracts with F-actin intact, almost all the ER, mitochondria, and acidic organelles remained evenly distributed over asters as they grew. A small fraction of the ER accumulated near MTOCs (*Figures 6A* and *7A,D*, *Videos 8* and *9*). The ER intensity around MTOCs increased to ~2-fold higher than the intensity outside the aster (*Figure 6A'*) in >5 examples scored. Although the majority of ER remained stationary, astral MTs did induce a subtle change in the texture of the ER, from coarser outside the aster, to finer and more tubular in appearance inside the aster (*Videos 8* and *9*). Astral MTs also affected the structure of the F-actin network, from random orientation of filaments outside the aster, to transient radial alignment of a subpopulation of bundles inside the aster (*Figure 7A*) as we reported previously (*Field et al., 2019*).

When F-actin was fragmented with Cytochalasin D, all organelles exhibited inward movement (*Figures 6B* and *7G*), which was fastest at the growing periphery of the aster (*Figure 7*; *Figure 7—figure supplement 1*). Compared to control, a greater fraction of the ER was transported inwards (*Figure 6B'*), and average transport speeds were an order of magnitude faster with F-actin fragmented than intact (*Figure 7*). The ER intensity around MTOCs accumulated to ~6 fold higher than the intensity outside the aster and continued to increase with time (*Figure 6B'*). Due to the burst of movement at the periphery of the growing aster, the intensity of organelles was ~30% lower there than outside the aster (*Figure 6B'*). Compared to control, the texture of

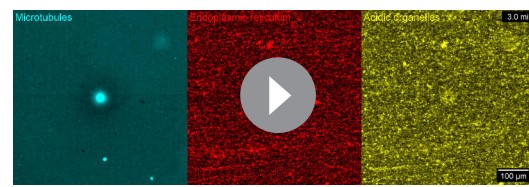

**Video 10.** Burst of ER and acidic organelle movement at the growing aster periphery with F-actin fragmented. (Related to *Figure 7*) Transport of ER and acidic organelles with F-actin fragmented by Cytochalasin D. MTs were labeled with tubulin-Alexa Fluor 647, ER with DiD, and acidic organelles with LysoTracker Red. Unlike in control with F-actin intact, the burst of movement near the aster periphery was highly reproducible when F-actin was fragmented with Cytochalasin D.
https://elifesciences.org/articles/60047#video10

the ER was coarser when F-actin was fragmented, both inside and outside asters, and MTs appeared more bundled. Mitochondria and acidic organelles moved inwards and accumulated near the MTOC. These organelles appeared to physically associate with ER in higher magnification images (*Figs 6B,C*, *7G*, *Figure 7—figure supplement 1*), so all organelles may be physically connected in this system. These findings show that the ER, and perhaps all organelles, recruit dynein, and can move toward the MTOC. Inward movement is restrained by F-actin under control conditions. However, even with F-actin fragmented, the majority of the ER, mitochondria, and acidic organelles were still evenly distributed over the aster.

We next added p150-CC1 to test for a role of dynein in organelle transport. With p150-CC1 present, with or without F-actin, organelles moved neither inwards nor outwards, and did not accumulate at MTOCs. This result is illustrated in *Figure 6C* for the p150-CC1 plus Cytochalasin D condition. We conclude that dynein generates the majority of force on organelles, and that other known forces, for example, from kinesins or tip tracking, do not induce significant net transport in our system, although they may drive transient saltatory motion.

## Dynein-mediated organelle movement is maximal near the aster periphery

To infer outward forces on MTs as a function of time and location, we needed a measure of the total inward organelle flux. Kymographs and PIV provide direct visualization of movement but have limitations for this inference, because they measure movement of local gradients in fluorescence intensity, not mass transport. We therefore developed an analysis to measure mass transport of organelles based on flux of fluorescence intensity (analysis described in *Figure 7—figure supplement 2* and Materials and methods). This analysis quantifies the amount of fluorescence signal crossing a given circumference at a given time, normalized by the total fluorescence in a region containing the aster. *Figure 7* shows examples with ER and acidic organelles. Mitochondria exhibited similar movement as acidic organelles (*Figure 7—figure supplement 1*).

All analysis methods revealed a burst of inward organelle movement when the growing aster periphery reached them, followed by slowing down inside asters (*Videos 9*, *10*, *11*). This burst can be visualized as inward diagonal features in kymographs, and red values on the diagonal corresponding to the growing aster periphery in mass transport and PIV plots. Under control conditions, with F-actin intact, the amount of organelle movement at the aster periphery was variable between extracts. Out of 11 extract preps, we observed a burst of inward ER movement at the aster periphery in seven extracts (64%) as in *Figure 7C*, and observed weaker or no burst in the remaining extracts as in *Figure 7F*. Factors that seem to lessen the burst of inward movement include higher concentrations of spontaneously nucleated MTs outside the aster, and insufficient passivation of the coverslips. Lack of fast organelle movement in control asters with intact F-actin is consistent with co-movement of cytoplasmic networks in moving asters (*Figures 3* and *4*).

When F-actin was fragmented, a burst of organelle transport at the growing aster periphery was observed in all experiments (>10 repeats

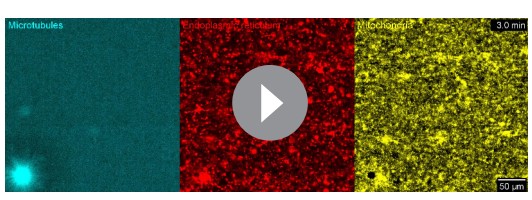

**Video 11.** Burst of ER and mitochondria movement at the growing aster periphery with F-actin fragmented. (Related to *Figure 7—figure supplement 1*) MTs were labeled with tubulin-Alexa Fluor 647, ER with DiD, and mitochondria with TMRE.
https://elifesciences.org/articles/60047#video11

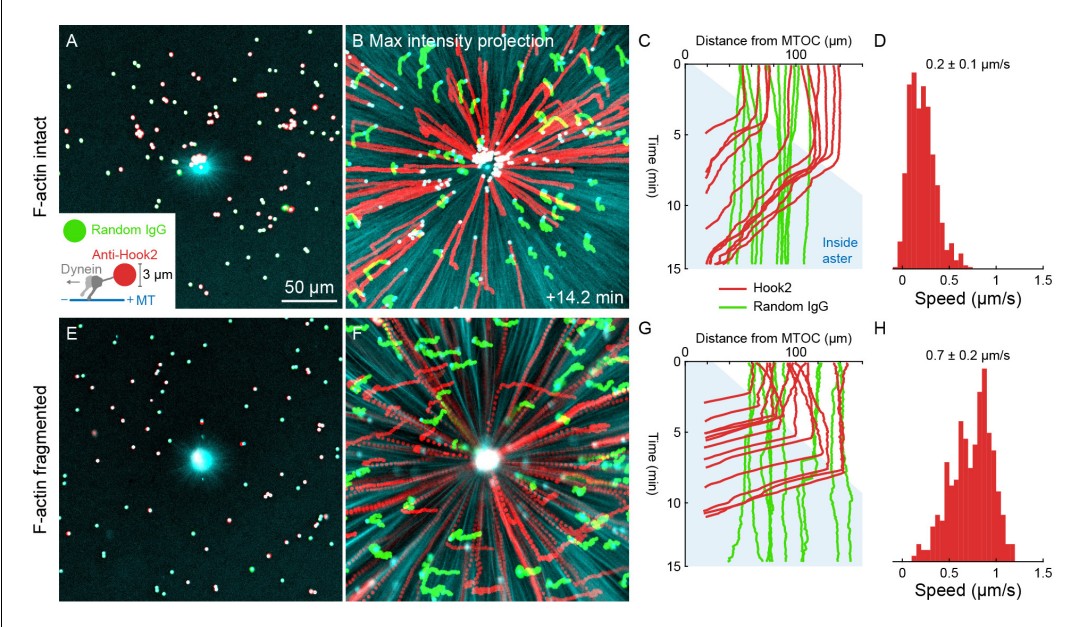

**Figure 8.** Unlike organelles, artificial cargoes functionalized with dynein move at constant speed throughout asters. (**A**) Artificial cargoes were functionalized with an antibody against the dynein adapter HOOK2, and negative control beads were functionalized with random antibody (see *Video 12*). (**B**) Max intensity projections of beads functionalized with anti-HOOK2 (red) or random antibody (green). (**C**) Trajectories of anti-HOOK2 and negative control beads relative to the MTOC. The growing aster is indicated by the blue region. Anti-HOOK2 beads started to be transported when they were engulfed by the growing aster. (**D**) Velocity distribution of anti-HOOK2 beads inside the aster. (**E–H**) Similar experiment with F-actin fragmented by Cytochalasin D.

The online version of this article includes the following source data for figure 8:

**Source data 1.** Bead trajectories for panels C, D, G, and H.

with different extracts). Inward movement at the periphery was faster than control conditions and therefore easier to visualize and quantify. Velocity values for ER moving inwards at the aster periphery reached ~0.25 µm/s with F-actin fragmented (*Figure 7J*), and mass transport reached 2% of total per min (*Figure 7I*). Mass transport values were more peaked at the aster periphery than PIV values, in part because mass transport takes into account the increase in circumference as the aster radius increases. A smaller fraction of acidic organelles than ER was transported inwards (*Figure 7F,I*), but with a similar bias toward more movement at the periphery. Although inward movement was faster with F-actin fragmented, it was still mostly confined to the periphery. More organelles accumulated at the aster center than in control (*Figure 6*), but most organelles were still uniformly spread over the aster and not moving, on average.

## Dynein-coated beads move inwards at constant rates throughout asters

Slowing of organelle transport upon incorporation into the aster suggested dynein might be inhibited inside asters. To test this, we turned to an artificial system. 2.8 µm diameter beads were functionalized with the antibody against the dynein adapter HOOK2 used in *Figures 4* and *5*. Negative control beads were functionalized with random IgG. We then measured transport of the beads on isolated, stationary asters as in *Figures 6* and *7*. With F-actin intact, the anti-HOOK2 beads moved inwards at a constant speed of 0.2 ± 0.1 µm/s throughout asters (*Figure 8A–D*, *Video 12*). When F-actin was fragmented with Cytochalasin D, the anti-HOOK2 beads moved at 0.7 ± 0.2 µm/s (*Figure 8E–H*), threefold faster than with F-actin intact. Thus, artificial dynein-coated beads were slowed by F-actin, like endogenous organelles. However, these beads were transported all the way to the MTOC, unlike organelles which slowed or stopped inside asters.

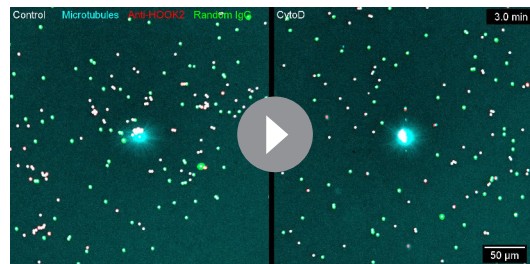

**Video 12.** Artificial cargoes, Dynabeads functionalized with dynein via anti-HOOK2, were transported at constant speeds throughout asters. (Related to *Figure 8*) MTs were labeled with tubulin-Alexa Fluor 488, anti-HOOK2 beads with Fab fragment-Alexa Fluor 568, and negative control beads were functionalized with random rabbit IgG and labeled with Fab fragment-Alexa Fluor 647.

https://elifesciences.org/articles/60047#video12

## Volume exclusion is unlikely to block organelle movement inside asters

Organelles might slow down inside asters because the environment becomes too crowded with other organelles. To investigate volume exclusion by organelles, we quantified the intensity and flux of fluorescent dextran as a marker for the cytosol. As organelles were transported toward MTOCs, fluorescent dextran was displaced away from MTOCs (*Figure 7—figure supplement 3*, *Video 13*), consistent with volume conservation. However, the degree of steric exclusion was fairly small, since the dextran signal was only reduced by ~10%, and exclusion was only observed within ~50 μm of MTOCs, where the ER density is maximal. Outside that central region, the intensity of fluorescent dextran was similar inside and outside asters. We conclude that volume exclusion between organelles may be significant in the immediate neighborhood of MTOCs, but is unlikely to account for organelles becoming stationary inside asters.

## Discussion

We tracked multiple cytoplasmic networks in moving asters using two different systems to promote movement, and found that the majority of organelles, F-actin, keratin, and even a small molecule probe moved coherently with astral MTs. Co-movement of cytoplasmic networks is consistent with mechanical integration between networks. MTs, F-actin and organelles mechanically interact via many motor and non-motor proteins, as reported in *Xenopus* egg extracts (*Lane and Allan, 1999*; *Waterman-Storer et al., 2000*; *Waterman-Storer et al., 1995*) and other systems (*Dogterom and Koenderink, 2019*; *Gurel et al., 2014*; *Mandato and Bement, 2003*; *Rodriguez et al., 2003*; *Semenova et al., 2008*; *Waterman-Storer and Salmon, 1998*). Many such proteins described in other systems are present in the *Xenopus* egg proteome (*Wühr et al., 2014*). Furthermore, nonspecific steric and hydrodynamic interactions may contribute to mechanical integration between the entangled cytoplasmic networks. Coherent movement of cytoplasmic networks has been reported in other systems where the entire cytoplasm is driven from the boundary, such as cytoplasmic flows with respect to the cortex in *Drosophila* embryos (*Deneke et al., 2019*), or rotation of *C. elegans* embryos with respect to the egg shell (*Schonegg et al., 2014*). Here, though the aster is self-driven within cytoplasm before interacting with the cortical boundary, inside the aster all components still moved together.

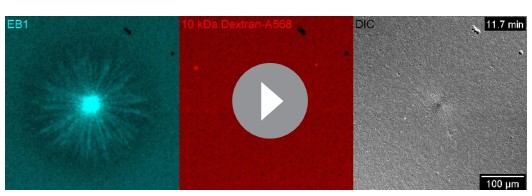

**Video 13.** Exclusion of 10 kDa dextran from the organelle-rich region around MTOCs. (Related to *Figure 7—figure supplement 3*) 10 kDa dextran labeled with Alexa Fluor 568 was excluded in a ~ 50 μm radius around the MTOC.

https://elifesciences.org/articles/60047#video13

Co-movement appears contradictory to many studies where organelles exhibit saltatory movement with respect to MTs, including in the egg extract system (*Lane and Allan, 1999*; *Wang et al., 2013*; *Waterman-Storer et al., 1995*). Most reviews of organelle systems assume they move with respect to MTs at rest. Ironically, the standard length-dependent pulling model of aster movement in eggs assumes the opposite, that MTs move with respect to organelles at rest. Co-movement of organelles with a moving cytoskeleton has been less studied. When we imaged at higher magnification, we too observed saltatory movement of organelles and F-actin with respect to astral MTs (*Figure 1—figure*

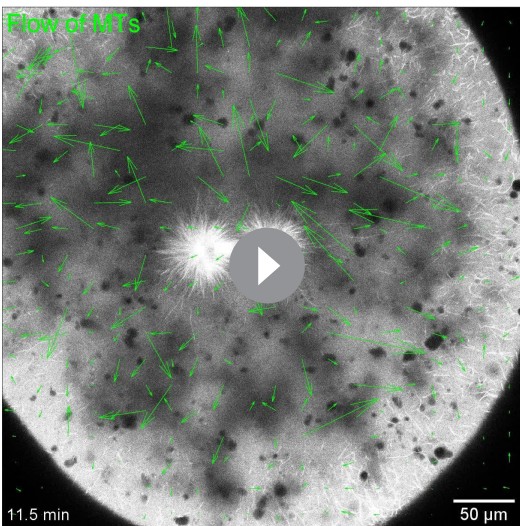

**Video 14.** Post-anaphase aster separation movement in a zebrafish embryo consistent with co-movement. (Related to *Figure 9*) Video from *Wühr et al., 2010* and analyzed with permission. Microtubules were labeled with microtubule-binding domain of Ensconsin fused to three GFPs (EMTB-3GFP) (*Faire et al., 1999*; *von Dassow et al., 2009*). Flows of MTs were estimated by PIV (Materials and methods).
https://elifesciences.org/articles/60047#video14

supplement 1) and transient deformation of the ER (*Figure 3—figure supplement 1*), consistent with previous reports. By imaging at lower magnification, we averaged movement over entire asters and were able to quantify the net fluxes on scales of hundreds of microns and tens of minutes. We believe these net fluxes are most relevant to physical models of aster movement. Co-movement may be especially relevant in large eggs, while relative movement may be more significant in smaller cells. We did observe dynein-mediated inward organelle movement relative to MTs over a distance of ~50 μm at the aster periphery (*Figure 7*). This distance corresponds to a relatively thin peripheral layer in frog egg asters, but it is larger than the cell radius in sea urchin or *C. elegans* eggs.

An important question is how well our extract aster movement systems model movement in eggs. After anaphase in *Xenopus* eggs, centrosomes move away from the midplane at ~10 μm/min, which is faster than the aster separation movement in *Figure 3*, and slower than the dynein-based movement over the coverslip in *Figure 4*. Thus, neither of our extract movement systems precisely reconstituted the speed of aster movement in eggs, but they spanned a wide range of relevant velocities. As a preliminary test of co-movement of MTs and organelles in living eggs, we re-analyzed videos of aster growth and separation movement after first mitosis in live zebrafish eggs expressing a fluorescent MT-binding protein (*Wühr et al., 2010*; *Video 14*). Lipid droplets are visible as large dark objects in these videos. These droplets move rapidly and randomly before the aster contacts them, then slowly outwards once they are embedded inside the aster. Using PIV analysis, we observed outward flow of structure in the MT channel at the same speed as the lipid droplets. This analysis suggests large asters in zebrafish eggs may also exhibit co-movement of MTs and organelles as they move apart after first mitosis.

Dynein located throughout the cytoplasm is thought to generate the force that moves asters in large egg cells, but the cytoplasmic cargoes to which dynein is anchored has been unclear. Here, we found that all the organelles in the extract can move inwards in a dynein-dependent manner, especially at the aster periphery. Thus, all the organelles may serve as dynein anchors, either by recruiting dynein directly, or by physical contact with the ER (*Guo et al., 2018*). The ER and mitochondria are the most abundant organelles, and the ER moved inwards fastest and to the greatest extent. Thus, ER may be the predominant dynein anchor in frog eggs. The identity of the dynein adapter on egg organelles is unknown. Eggs contain abundant lipid droplets and yolk platelets that are removed during extract preparation and could constitute additional dynein anchors.

Organelles reproducibly exhibited a burst of inward movement when the growing aster periphery first contacted them, then slowed or halted upon incorporation into the aster, as shown in both mass transport and PIV analyses (*Figures 6* and *7*). Most organelles inside asters were stationary, which explains why the density of organelles in the bulk of the aster was similar to that outside the aster, as previously observed in egg extracts (*Hara and Merten, 2015*; *Wang et al., 2013*). The same is true in intact eggs (*Figure 1B',C'*). The molecular mechanism that slows dynein-mediated movement of organelles inside asters is unknown. F-actin decreased dynein-based transport of both organelles and dynein-coated beads, so it is partly responsible. Organelle transport, but not bead transport, slowed inside asters even when F-actin was fragmented. We hypothesize that non-dynein interactions between organelles and MTs cause a braking effect. Candidate brakes include opposing motors, tip tracking factors, non-motor 'brake' proteins, and nonspecific steric or hydrodynamic

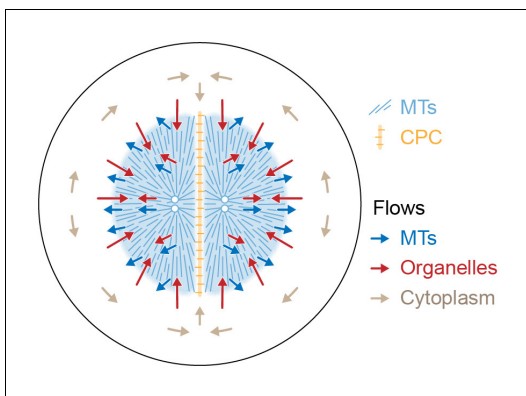

**Figure 9.** Model for component flows during aster separation movement in frog eggs. Within moving asters, all cytoplasmic networks move together, advecting cytosol. Near the aster periphery, organelles flow rapidly inwards while MTs flow outwards. Outside the aster, cytosol is displaced around asters and into the midplane by hydrodynamic forces.

interactions. We did not observe significant mass transport away from MTOCs by kinesins or tip tracking when dynein was inhibited, which argues against opposing motors. The predominance of dynein as the organelle motor in *Xenopus* eggs is consistent with previous studies using high-magnification DIC imaging (*Lane and Allan, 1999*). ER transport by kinesin increases as the embryo develops (*Lane and Allan, 1999*).

*Figure 9* proposes a model for the flows associated with aster separation movement in eggs. At the growing aster periphery, organelles flow inwards rapidly but transiently, while astral MTs both grow and flow outwards. In the aster interior, organelles and MTs flow outwards together. Moving asters advect cytosol, generating hydrodynamic forces that displace cytoplasm around the asters and into the midplane (beige arrows in *Figure 9*). This convective flow supplies components to support aster growth at the midplane boundary. F-actin is not shown, but we believe it slows organelle movement relative to MTs throughout the aster and disassembles at the midplane, generating a spatial asymmetry that helps asters move apart. The new data in this paper report on flows, not forces, so we cannot immediately extrapolate from *Figure 9* to a force model. However, the flows in *Figure 9* are incompatible with the classic length-dependent force model in its simplest form, where the MT component of the aster is conceptualized as a rigid body moving through a viscous cytoplasm (*Hamaguchi and Hiramoto, 1986*; *Palenzuela et al., 2020*; *Sallé et al., 2019*; *Tanimoto et al., 2016*; *Tanimoto et al., 2018*). Our observations suggest that in frog eggs, where large asters are built from a network of short MTs entangled with ER and F-actin, the aster is better conceptualized as a deformable gel where the midplane is softer than the aster interior. In this framework, MTOCs could move in response to forces exerted on the surface of the gel, where vesicles move relative to MTs, and perhaps also in response to stresses within it. Inside the aster, organelles appear to interact with MTs using both dynein motors and unidentified brakes. Dynein on organelles anchored to the MT network could generate active stresses that deform the gel. We are not the first to propose that asters behave as a mechanically integrated gel. In early microneedle experiments in echinoderm eggs, *Chambers, 1917* observed that asters behave as a gel and proposed that forces act on their surface. How surface and internal forces and stresses contribute to MTOC movement, and how each scales with aster size and shape, are interesting topics for further research.

## Materials and methods

### Key resources table

| Reagent type (species) or resource | Designation | Source or reference | Identifiers | Additional information |
|---|---|---|---|---|
| Strain, strain background (*Escherichia coli*) | Rosetta 2(DE3)pLysS competent cells | Novagen | Cat#: 71401 | Competent cells |
| Strain, strain background (*Escherichia coli*) | BL21(DE3)pLysS competent cells | Promega | Cat#: L1195 | Competent cells |
| Biological sample (*Xenopus laevis* adult females) | Eggs | Harvard Medical School *Xenopus* Colony | | http://www.xenbase.org/entry/ |

*Continued on next page*

*Continued*

| Reagent type (species) or resource | Designation | Source or reference | Identifiers | Additional information |
|---|---|---|---|---|
| Biological sample (*Xenopus laevis* adult females) | Egg extracts | *Field et al., 2017* | | |
| Biological sample (*Bos taurus*) | Fluorescently labeled tubulin from bovine brain | *Desai and Mitchison, 1998*; *Miller and Wilson, 2010* | | |
| Biological sample (*Xenopus laevis* adult females) | Fluorescently labeled tubulin from frog egg extract | *Groen and Mitchison, 2016* | | |
| Antibody | Anti-tubulin, clone B-5-1-2. (Mouse monoclonal) | Sigma-Aldrich | Cat#: T6074 RRID:AB_477582 | IHC (1:2000) |
| Antibody | Anti-*Xenopus* LNPK, raised against cytosolic fragment of LNPK, aa 99–441. (Rabbit polyclonal) | *Wang et al., 2016* | | IHC (1:1000) |
| Antibody | Anti-PDIA3 (Rabbit polyclonal) | Boster Bio | Cat#: PB9772 | IHC (1:1000) |
| Antibody | Anti-*Xenopus* AURKA, raised against full-length protein (Rabbit polyclonal) | *Field et al., 2017*; *Tsai and Zheng, 2005* | | IP (3 µL per 50 µL Dynabeads, to saturate Dynabeads) |
| Antibody | Anti-INCENP, C-terminal peptide immunogen. CAVWHSPPLSSNRHHLAVGYGLKY (Rabbit polyclonal) | *Sampath et al., 2004* | | IHC (1:1200) |
| Antibody | Anti-cytokeratin pan, clone C-11 (Mouse monoclonal) | Sigma-Aldrich | Cat#: P2871 RRID:AB_261980 | IHC (1:500) |
| Antibody | Anti-HOOK2, C-terminal peptide immunogen. CSRSHTLLPRYTDKRQSLS (Rabbit polyclonal) | This paper | | See Materials and methods, 'HOOK2 antibody' and 'Preparation of dynein on coverslips' |
| Antibody | ChromPure rabbit IgG, whole molecule | Jackson Immuno Research | Cat#: 011-000-003 RRID:AB_2337118 | IP (1 µL per 50 µL Dynabeads, to saturate Dynabeads) |
| Antibody | Goat anti-rabbit whole serum | Jackson Immuno Research | Cat#: 111-001-001 RRID:AB_2337909 | See Materials and methods, 'Photo-release of fluorescein from MTOCs' |
| Peptide, recombinant protein | Tau MT-binding domain (mTMBD)-mCherry, *E. coli* expression | *Mooney et al., 2017* | | |
| Peptide, recombinant protein | EB1-GFP, *E. coli* expression | *Nguyen et al., 2014* | | |
| Peptide, recombinant protein | Lifeact-GFP, *E. coli* expression | *Moorhouse et al., 2015*; *Riedl et al., 2008* | | |
| Peptide, recombinant protein | p150-CC1 fragment of dynactin, *E. coli* expression | *King et al., 2003* | | |
| Peptide, recombinant protein | NeutrAvidin | Thermo Fisher | Cat#: 31000 | |
| Peptide, recombinant protein | Biotinylated Protein A | GenScript | Cat#: M00095 | |
| Commercial assay or kit | 2.8 µm Protein A coated Dynabeads | Thermo Fisher | Cat#: 10002D | |
| Commercial assay or kit | 2.8 µm Protein G coated Dynabeads | Thermo Fisher | Cat#: 10004D | |

*Continued on next page*

Continued

| Reagent type (species) or resource | Designation | Source or reference | Identifiers | Additional information |
|---|---|---|---|---|
| Commercial assay or kit | Protein G UltraLink resin | Thermo Fisher | Cat#: 53125 | |
| Commercial assay or kit | Affi-Prep Protein A resin | Bio-Rad | Cat#: 1560006 | |
| Commercial assay or kit | HisPur cobalt resin | Thermo Fisher | Cat#: 89965 | |
| Commercial assay or kit | Superdex Increase 75 10/300 GL column | GE Healthcare | Cat#: 29-1487-21 | |
| Chemical compound, drug | Formamide | Sigma-Aldrich | Cat#: F9037 | |
| Chemical compound, drug | Benzyl benzoate | Sigma-Aldrich | Cat#: B6630 | |
| Chemical compound, drug | Benzyl alcohol | Sigma-Aldrich | Cat#: 402834 | |
| Chemical compound, drug | IGEPAL CA-630 | Sigma-Aldrich | Cat#: I8896 | |
| Chemical compound, drug | DiI (1,1'-dioctadecyl-3,3,3',3'-tetramethylindocarbocyanine perchlorate, aka DiIC18(3)) | Thermo Fisher | Cat#: D282 | |
| Chemical compound, drug | DiD (1,1'-dioctadecyl-3,3,3',3'-tetramethylindodicarbocyanine perchlorate, aka DiIC18(5)) | Thermo Fisher | Cat#: D307 | |
| Chemical compound, drug | Tetramethylrhodamine, ethyl ester (TMRE) | Thermo Fisher | Cat#: T669 | |
| Chemical compound, drug | LysoTracker Red DND-99 | Thermo Fisher | Cat#: L7528 | |
| Chemical compound, drug | Cytochalasin D | Cayman Chemical | Cat#: 11330 | |
| Chemical compound, drug | Poly-L-lysine-g-polyethylene glycol (PLL-g-PEG) | SuSoS Chemicals | Cat#: [PLL(20)-g[3.5]-PEG(2)] | |
| Chemical compound, drug | Lanolin for VALAP (Vaseline, lanolin, paraffin 1:1:1 by mass) | Sigma-Aldrich | Cat#: L7387 | |
| Chemical compound, drug | Paraffin | Sigma-Aldrich | Cat#: 327204 | |
| Chemical compound, drug | Phenylmethylsulfonyl fluoride (PMSF) | Sigma-Aldrich | Cat#: 78830 | |
| Chemical compound, drug | AlexaFluor-488,–568, –647 NHS ester | Thermo Fisher | Cat#: A20100, Cat#: A20003, Cat#: A20106 | |
| Chemical compound, drug | Caged fluorescein | *Mitchison et al., 1998* | | |
| Chemical compound, drug | 1-Ethyl-3-(3-dimethylaminopropyl) carbodiimide (EDC) | Thermo Fisher | Cat#: 22980 | |
| Software, algorithm | Fiji | *Schindelin et al., 2012* | RRID:SCR_002285 | |
| Software, algorithm | PIVlab | *Thielicke and Stamhuis, 2014* | | |
| Software, algorithm | 2D Gaussian fitting in MATLAB | *Nootz, 2020* | | |
| Software, algorithm | Radial mass transport analysis in MATLAB | This paper | | See Materials and methods, 'Analysis of organelle mass transport' |
| Other | Extended Liner Tape, thickness 25 μm, for flow cells | 3M | Cat#: 920XL | |

## Immunofluorescence

Embryos were fixed and stained as described previously (*Field et al., 2019*). Embryos were fixed in 90% methanol, 10% water, 50 mM EGTA pH 6.8 for 24 hr at room temperature with gentle shaking. After fixation, embryos were rehydrated in steps from 75%, 50%, 25%, to 0% methanol in TBS (50 mM Tris pH 7.5, 150 mM NaCl) for 15 min each step with gentle shaking. Rehydrated embryos in TBS were cut in half on an agarose cushion using a small razor blade. Before staining, embryos were bleached overnight in 1% hydrogen peroxide, 5% formamide (Sigma-Aldrich #F9037), 0.5x SSC (75 mM NaCl, 8 mM sodium citrate pH 7). To stain, embryos were incubated with directly labeled anti-bodies at 0.5–2 µg/mL for at least 24 hr at 4°C with very gentle rotation. Antibodies were diluted in TBSN (10 mM Tris-Cl pH 7.4, 155 mM NaCl, 1% IGEPAL CA-630 (Sigma-Aldrich #I8896), 1% BSA, 2% FCS, 0.1% sodium azide). After antibody incubation, embryos were washed in TBSN for at least 48 hr with several solution changes, then washed once in TBS and twice in methanol, with methanol washes for 40 min each. Embryos were cleared in Murray clear solution (benzyl benzoate (Sigma-Aldrich #B6630)/benzyl alcohol (Sigma-Aldrich #402834) 2:1). Embryos were mounted in metal slides 1.2 mm thick with a hole in the center. The hole was closed by sealing a coverslip to the bottom of the slide using heated Parafilm.

Endoplasmic reticulum (ER) was labeled with an anti-LNPK antibody (*Wang et al., 2016*) directly labeled with Alexa Fluor 568 NHS ester (Thermo Fisher #A20003). The ER was also probed with labeled anti-Protein disulfide-isomerase A3 (PDIA3) (Boster Bio #PB9772). PDIA3 is an ER lumen protein and had a similar distribution as the anti-LNPK antibody (not shown). MTs were labeled with an anti-tubulin antibody (Sigma-Aldrich #T6074, RRID:AB_477582) directly labeled with Alexa Fluor 647 NHS ester.

## Extract preparations

Actin-intact, CSF *Xenopus* egg extract was prepared as described previously (*Field et al., 2017*). CSF extracts were stored at 4–10°C and flicked occasionally to disperse membranes. Extracts stored in this way were typically usable for ~8 hr. Before each reaction, extracts were cooled on ice to ensure depolymerization of cytoskeletal networks.

## Interphase aster assembly reactions

In a typical reaction, fluorescent probes were added to CSF extract on ice. To trigger exit from CSF arrest and entry to interphase, calcium chloride was added to 0.4 mM final concentration. To ensure complete progression to interphase, the reaction was mixed well immediately after calcium addition by gently flicking and pipetting. Extracts were pipetted using 200 µL pipette tips manually cut to a wider bore to reduce shear damage, which can make membranes in the extract appear coarser by eye. Reactions were incubated in an 18°C water bath for 5 min then returned to ice for 3 min. Next, drugs or dominant negative constructs were added (see Perturbations below), and in some cases reactions were split for direct comparison between control and perturbed conditions. Last, Dyna-beads Protein G (Thermo Fisher #10004D) functionalized with an activating anti-Aurora kinase A (anti-AURKA) antibody were added as artificial microtubule organizing centers (MTOCs) (*Tsai and Zheng, 2005*). For experiments in which asters moved away from one another, unlabeled anti-INCENP antibody was included at a final concentration of 4 nM to promote zone formation by activating the CPC.

## Coverslip passivation

Eighteen and 22 mm square coverslips were passivated with poly-L-lysine covalently grafted to poly-ethylene glycol (PLL-g-PEG) (SuSoS #PLL(20)-g3.5-PEG(2)) as described previously (*Field et al., 2019*). Coverslips were cleaned by dipping them in 70% ethanol, igniting the ethanol with a gas burner, cooling the coverslips for several seconds, then the coverslips were passivated by placing them on a droplet of 0.1 mg/mL PLL-g-PEG in 10 mM HEPES pH 7.4 on Parafilm. Eighteen mm coverslips were placed on 90 µL droplets, and 22 mm coverslips were placed on 110 µL droplets. After 30 min incubation, excess PLL-g-PEG was rinsed by placing coverslips on droplets of distilled water twice for 5 min each, then drying them with a stream of nitrogen gas. To check the passivation, when we focused near the coverslips, we found no evidence of a surface layer of cytoskeletal filaments or organelles adsorbed to the coverslips. Quite the opposite, the density of cytoplasmic

networks was typically lower near the coverslips and higher near the midplane between the coverslips, we suspect due to continuous contraction of actomyosin away from the coverslips sustained by continuous diffusion of monomer toward the coverslips.

## Flow cell assembly

Flow cells were assembled from the passivated coverslips to increase physical stability of the system and reduce global flows. To a metal slide holder, 22 mm square coverslips were sealed via a thin layer of molten VALAP (Vaseline, lanolin (Sigma-Aldrich #L7387), paraffin (Sigma-Aldrich # 327204) 1:1:1 by mass). Then an 18 mm square coverslip was immobilized above the 22 mm coverslip using two pieces of thin double-sided tape (3M Extended Liner Tape #920XL) spaced ~1 cm apart. The tape has a nominal thickness of 25 µm and resulted in flow cells ~20 µm deep after pressing the coverslips together.

## Imaging

Extract reactions were perfused into flow cells, then the edges were sealed with VALAP. In experiments with a single condition, imaging was started immediately. In experiments with multiple conditions imaged in parallel, the slide holder was first chilled on ice for several seconds, so aster growth would start at the same time across conditions. Extracts were imaged on a Nikon Eclipse Ti2-E inverted microscope with Nikon CFI Plan Apo Lambda 20x, NA 0.75 objective lens, SOLA SE V-nIR light engine, and with either a Nikon DS-Qi2 or Andor Zyla 4.2 PLUS sCMOS camera. The microscope room was cooled to less than 20°C, otherwise spontaneously nucleated MTs can overtake reactions. Throughout the paper, time is measured with respect to warming the reaction and the start of aster growth. Depending on the MTOC density, asters typically grew into contact at 8–15 min and formed CPC-positive interaction zones several minutes later.

## Fluorescent probes

MTs were imaged with either bovine or frog tubulin directly labeled with Alexa Fluor 647 at a final concentration of 250 nM, or with a phosphodeficient version of the MT-binding domain of Tau fused to mCherry (*Mooney et al., 2017*) at a final concentration of 20 nM. Growing +TIPs of MTs were labeled with EB1-GFP at a final concentration of 110 nM. The chromosomal passenger complex (CPC) was labeled with an anti-INCENP antibody directly labeled with Alexa Fluor 647 at a final concentration of 4 nM. ER was labeled with DiI (1,1′-dioctadecyl-3,3,3′,3′-tetramethylindocarbocyanine perchlorate, aka $DiIC_{18}(3)$) (Thermo Fisher #D282) or DiD (1,1′-dioctadecyl-3,3,3′,3′-tetramethylindo-dicarbocyanine perchlorate, aka $DiIC_{18}(5)$) (Thermo Fisher #D307) at a final concentration of 4 µg/mL. Mitochondria were labeled with tetramethylrhodamine, ethyl ester (TMRE) (Thermo Fisher #T669) at a final concentration of 0.3 µg/mL. Acidic organelles were labeled with LysoTracker Red DND-99 (Thermo Fisher #L7528) at a final concentration of 130 nM. To allow these dyes to pre-incorporate into the membranous organelles, especially important for the DiI and DiD, stock solutions were first dissolved in DMSO to a concentration of 2 mg/mL (DiI/DiD), 0.2 mg/mL (TMRE), or 200 µM (LysoTracker Red), then diluted 50 fold into extract. These extract working solutions were incubated in an 18°C water bath for 45 min, flicking every 15 min to disperse membranes. Then the extract working solutions were stored on ice until use, then diluted an additional 10–30 fold into the final reaction. F-actin was imaged with Lifeact-GFP (*Moorhouse et al., 2015*; *Riedl et al., 2008*) at a final concentration of 300 nM. More details on fluorescent probes are reported in *Field et al., 2017*. Keratin was imaged with an anti-cytokeratin antibody (Sigma-Aldrich #P2871, RRID:AB_261980) directly labeled with Alexa Fluor 568 at a final concentration of 3 µg/mL.

## Perturbations

To fragment F-actin, Cytochalasin D (CytoD) was added to a final concentration of 20 µg/mL. CytoD was diluted in DMSO to 10 mg/mL, then diluted 20-fold into extract. This extract working solution was stored on ice until use, then diluted an additional 25 fold into the final reaction. CytoD and other drugs or dominant negative constructs were typically added to actin-intact extracts after cycling to interphase, then reactions were split for direct comparison between control and perturbed extracts. Alternatively, CytoD may be added during extract preparations before the crushing spin, following the classic CSF extract protocol (*Murray, 1991*). The ER appeared coarser in CytoD

extracts than in actin-intact extracts, and the ER appeared to coarsen over time in actin-intact extracts plus CytoD.

To inhibit dynein, the p150-CC1 fragment of dynactin (*King et al., 2003*), which acts as a dominant negative for dynein function, was added to a final concentration of 40 µg/mL.

## HOOK2 antibody

An affinity-purified C-terminal peptide antibody was produced in rabbit against *Xenopus laevis* HOOK2 (C-SRSHTLLPRYTDKRQSLS) (Cocalico Biologicals, Inc, PA).

## HOOK2 immunoprecipitation-mass spectrometry (IP-MS)

Dynabeads Protein G (Thermo Fisher #10004D) (20 µL Dynabeads slurry per reaction) were saturated with rabbit IgG (anti-HOOK2 or random IgG; Jackson ImmunoResearch #011-000-003, RRID:AB_ 2337118) by overnight binding, then washed 3x with CSF-XB (100 mM KCl, 2 mM MgCl2, 0.1 mM CaCl2, 10 mM K HEPES pH 7.7, 5 mM EGTA, 50 mM sucrose). Each immunoprecipitation reaction contained 150 µL interphase or CSF-arrested egg extract treated with 10 µg/mL Cytochalasin D to inhibit gelation. Extract plus Dynabeads was rotated gently for 60 min at 4°C, then washed 4x in 50 mM KCl, 1 mM MgCl2, 10 mM K HEPES pH 7.7, 1 mM EGTA at 0°C. The tubes were changed twice during the washes to remove extract protein bound to their walls. Protein bound to the Dynabeads was eluted in 20 µL of 5 M guanidine thiocyanate, 5 mM dithiothreitol (DTT) (US Biological #D8070) for 10 min at 60°C, then cysteines were alkylated with N-ethylmaleimide (NEM). The eluate was precipitated with chloroform-methanol then subjected to proteolysis followed by TMT labeling as described (*Sonnett et al., 2018*).

## Preparation of dynein on coverslips

Coverslips were passivated following the protocol above but using biotinylated PLL-g-PEG. NeutrAvidin (Thermo Fisher #31000) and biotinylated Protein A (GenScript #M00095) were mixed in a 1:1 ratio to a final concentration of 10 µM and stored at 4°C. Just before functionalizing the coverslips, the NeutrAvidin and biotinylated Protein A mixture was diluted 42-fold to 240 nM in 1x PBS with 0.0025% Tween 20. That concentration was found to be the smallest amount to decrease the surface tension enough to maintain a layer of solution on the coverslips, to reduce damage to the functionalized surfaces due to air-water interfaces when transferring the coverslips from one droplet to another. Coverslips were incubated with the NeutrAvidin and biotinylated protein A mixture at least 30 min on droplets on Parafilm at room temperature. Coverslips were incubated under a box with a damp paper towel, to block room light and to reduce evaporation. After the incubation, coverslips were rinsed twice on droplets of 1x PBS with 0.0025% Tween 20 for 5 min each, then incubated with anti-HOOK2 or random IgG diluted in 1x PBS with 0.0025% to a final concentration of 10 µg/mL at least 30 min. After the incubation with antibody, coverslips were rinsed twice on droplets of 1x PBS with 0.0025% Tween 20, then twice on droplets of distilled water, then swirled in a beaker of distilled water, then gently dried with a stream of nitrogen gas. Coverslips were often used same day, but could be stored overnight in the dark at 4°C and used the following day. After perfusing extracts into flow cells and sealing the edges with VALAP, the metal slide holders were chilled for 10 min on a metal block on ice, to allow endogenous HOOK2 and dynein-dynactin time to bind the anti-HOOK2 before the start of aster growth.

## Photo-release of fluorescein from MTOCs

Caged fluorescein with -O-CH$_2$-COOH functionality on the caging groups was synthesized as described (*Mitchison et al., 1998*). Carboxylic acid groups were activated as sulfo-NHS esters in a small reaction containing 2 micromols caged fluorescein, 5 micromols sodium sulfo-NHS and 5 micromols 1-ethyl-3-(3-dimethylaminopropyl)carbodiimide (EDC) (Thermo Fisher #22980) in 10 µL of DMSO. After 1 hr at room temperature, this reaction mix was added directly to protein coated beads. Direct modification of anti-AURKA beads caused loss of nucleation activity, so we first biotinylated beads, then modified with caged fluorescein, then attached anti-AURKA IgG using a NeutrAvidin bridge. In particular, Dynabeads Protein A (Thermo Fisher #10002D) were sequentially incubated with goat anti-rabbit whole serum (Jackson ImmunoResearch #111-001-001, RRID:AB_ 2337909) then biotinylated rabbit IgG (homemade). They were labeled with the caged fluorescein

reaction mix in 0.1 M K HEPES pH 7.7 for 1 hr, then washed again. We empirically titrated the amount of reaction mix added such that beads were maximally labeled while still retaining nucleation activity in extract. After labeling with caged fluorescein, beads were incubated sequentially with a mixture of NeutrAvidin and biotinylated protein A, then rabbit anti-AURKA to confer nucleation activity. Pure proteins were added at 10–20 µg/mL and serum was added at 1/20. All binding reactions were incubated for 20 min, and washes were in 1x PBS. Fluorescein was released from beads by exposing the microscope field to full illumination in the DAPI channel (395 nm) for 5 s.

## Analyses

### PIV

PIVlab (*Thielicke and Stamhuis, 2014*) was used to estimate flow fields of cytoplasmic networks based on particle image velocimetry (PIV). Though PIV is primarily used to estimate flow fields based on tracer particles embedded in fluids, PIV has been used to estimate cortical or cytoplasmic flows in *C. elegans* cortices (*Mayer et al., 2010*), zebrafish epithelia (*Behrndt et al., 2012*), and *Drosophila* embryos (*Deneke et al., 2019*). Likewise, cytoplasmic networks in the *Xenopus* egg extracts included structures with sufficient contrast for PIV. The cytoplasmic networks exhibited dynamic turnover, so it was important to image with a time interval short enough to retain sufficient correlation between frames for PIV. For example, the time scale for F-actin turnover was ~1 min, based on recovery of F-actin in a region where F-actin had been mechanically cleared, consistent with estimates based on measurements of network density and flow in contractile actomyosin networks (*Malik-Garbi et al., 2019*). Time intervals less than 20 s worked well for PIV.

### Gaussian fitting of photo-released fluorescein

2D Gaussian fitting of fluorescein photo-released from MTOCs was performed using a nonlinear least squares solver in MATLAB (*Nootz, 2020*). After photo-release the MTOCs were bright due to uncaged fluorescein that remained bound to the MTOCs. Thus the MTOCs were masked as not to bias the Gaussian fits. Positional error associated with Gaussian fitting is $2/\pi(h\mu/A)^2$ (*Condon, 1997*), where $h = 0.32$ is the pixel size, $\mu \approx 20$ is the standard deviation of the noise, and $A \approx 10$ is the peak amplitude. With these values, the positional error is on the order of a pixel, which is significantly smaller than the difference between the bead position and the center of the cloud. We fit expansion of the fluorescein cloud to a model of diffusion, and we assumed a diffusion coefficient of fluorescein in water of 425 µm²/s (*Culbertson et al., 2002*). Advection of cytosol with cytoplasmic networks is consistent with a poroelastic Péclet number $VL\mu/E\xi^2$ greater than unity (*Mitchison et al., 2008*; *Moeendarbary et al., 2013*). Given the oscillatory speed $V \sim 1$ µm/s (*Figure 4D*) and amplitude $L \sim 30$ µm (*Figure 4C*), and assuming a viscosity $\mu \sim 6$ x water (*Figure 5E*) and an elastic modulus $E \sim 10$ Pa (*Valentine et al., 2005*), we estimate the upper bound on the effective pore size $\xi$ of cytoplasmic networks in this system is ~100 nm.

### Analysis of organelle mass transport

The flux-based analysis of organelle transport is described in *Figure 7—figure supplement 2*. In summary, images were background subtracted and flat field corrected, then a region of interest (ROI) was defined large enough to enclose the aster at all time points, so the total amount of ER in the ROI was conserved. Then, the total intensity was normalized across frames to correct for photobleaching. The net flux of organelle fluorescence intensity toward MTOCs was calculated as described in *Figure 7—figure supplement 2*. In particular, the average intensity was calculated in annular bins with a width of 10 µm, then the cumulative total intensity was calculated from the MTOC to outside the aster, then the net flux was calculated at each radial distance by subtracting subsequent cumulative total intensity profiles.

## Acknowledgements

This work was supported by NIH grant R35GM131753 (TJM) and MBL fellowships from the Evans Foundation, MBL Associates, and the Colwin Fund (TJM and CMF). JFP was supported by the Fannie and John Hertz Foundation, the Fakhri lab at MIT, the MIT Department of Physics, and the MIT Center for Bits and Atoms. The authors thank the Nikon Imaging Center at Harvard Medical School and

Nikon at MBL for imaging support, and the National *Xenopus* Resource at MBL for support. The authors thank Keisuke Ishihara and Luolan Bai for critical feedback on the manuscript, and thank Martin Wühr, Jay Gatlin, Nikta Fakhri, David Burgess, Fabian Romano-Chernac, Sam Reck-Peterson, and Mark Terasaki for helpful conversations. The authors thank Martin Wühr for the video showing post-anaphase separation movement of asters after first mitosis in zebrafish. The anti-LNPK antibody was a gift from Tom Rapoport (Harvard Medical School and Howard Hughes Medical Institute, Boston, MA). The Tau-mCherry construct was a gift from Jay Gatlin (University of Wyoming, Laramie, WY). The EB1-GFP construct was a gift from Kevin Slep (UNC Chapel Hill, NC). The Lifeact-GFP construct was a gift from David Burgess (Boston College, Newton, MA).

## Additional information

### Funding

| Funder | Grant reference number | Author |
|---|---|---|
| National Institute of General Medical Sciences | R35GM131753 | Timothy J Mitchison |
| Hertz Foundation | | James F Pelletier |
| Ed Evans Foundation | MBL fellowships | Christine M Field Timothy J Mitchison |

The funders had no role in study design, data collection and interpretation, or the decision to submit the work for publication.

### Author contributions

James F Pelletier, Conceptualization, Formal analysis, Investigation, Writing - original draft; Christine M Field, Timothy J Mitchison, Conceptualization, Investigation, Writing - original draft; Sebastian Fürthauer, Formal analysis, Writing - review and editing; Matthew Sonnett, Resources, Helped with sample preparations and performed mass spectrometry (MS) measurements for Fig 4-figure supplement 1

### Author ORCIDs

James F Pelletier (iD) https://orcid.org/0000-0001-8064-0259
Christine M Field (iD) https://orcid.org/0000-0002-6433-7804
Sebastian Fürthauer (iD) https://orcid.org/0000-0001-9581-5963
Timothy J Mitchison (iD) https://orcid.org/0000-0001-7781-1897

### Ethics

Animal experimentation: This study was performed in strict accordance with the recommendations in the Guide for the Care and Use of Laboratory Animals of the National Institutes of Health. All of the animals were handled according to approved institutional animal care and use committee (IACUC) protocols (#IS00000519-3) of Harvard Medical School. The institution has an approved Animal Welfare Assurance on file with the Office of Laboratory Animal Welfare. The Assurance number on file is #D16-00270.

### Decision letter and Author response

Decision letter https://doi.org/10.7554/eLife.60047.sa1
Author response https://doi.org/10.7554/eLife.60047.sa2

## Additional files

### Supplementary files

- Transparent reporting form

## Data availability

All data generated or analyzed during this study are included in the manuscript and supporting files, and related code has been uploaded to GitHub: https://github.com/jamespelletier/Co-movement (copy archived at https://archive.softwareheritage.org/swh:1:rev:8144aa215bad15e091e267fc2-ba247ddc1c1db2d/).

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
