## [Decision Letter]

**Acceptance summary:**

This study reports elegant quantitative measurements demonstrating that in *Xenopus* egg cytoplasm filament and organelle networks are to a large extent mechanically integrated. This observation is important with respect to proposed models of the mechanism responsible for moving microtubule asters apart after anaphase of mitosis in large embryonic cells.

**Decision letter after peer review:**

Thank you for submitting your article "Co-movement of astral microtubules, organelles and F-actin suggests aster positioning by surface forces in frog eggs" for consideration by *eLife*. Your article has been reviewed by three peer reviewers, and the evaluation has been overseen by a Reviewing Editor and Vivek Malhotra as the Senior Editor. The following individual involved in review of your submission has agreed to reveal their identity: Stefano Di Talia (Reviewer #1).

The reviewers have discussed the reviews with one another and the Reviewing Editor has drafted this decision to help you prepare a revised submission.

As the editors have judged that your manuscript is of interest, but as described below that additional experiments are required before it is published, we would like to draw your attention to changes in our revision policy that we have made in response to COVID-19 (https://elifesciences.org/articles/57162). First, because many researchers have temporarily lost or reduced access to the labs, we will give authors as much time as they need to submit revised manuscripts. We are also offering, if you choose, to post the manuscript to bioRxiv (if it is not already there) along with this decision letter and a formal designation that the manuscript is "in revision at *eLife*". Please let us know if you would like to pursue this option. (If your work is more suitable for medRxiv, you will need to post the preprint yourself, as the mechanisms for us to do so are still in development.)

Summary:

This study aims to understand the driving forces responsible for microtubule aster movement in *Xenopus* eggs. A prominent model for aster movement in large cells, where microtubules do not contact the cell cortex, is that dynein-attached organelle movement generates a net drag-force that moves the aster in the opposite direction. Here the authors test this model by imaging the dynamics of aster separation in *Xenopus* egg extract. They show for the first time that asters, actin filaments, ER and bulk cytoplasm move together in the same direction, contrary to what a prevalent model predicts. Conversely, they find subtle but significant organelle movement at the "surface of the aster" where microtubules grow and penetrate the cytoplasm. Therefore, they propose a modified model which posits that vesicle movements exclusively in the aster periphery, instead of everywhere within the aster, drive aster movement in extract. While all reviewers agreed that this manuscript presents new and very interesting observations that argue against the "length-dependent pulling model" in egg extract, most reviewers concluded that a coherent biophysical picture explaining the observed aster movements is still missing. Specifically, it remains unclear if the source of the forces is related to the vesicle movement at the periphery of the asters.

Essential revisions:

1) The authors need to provide a clearer biophysical picture supporting their model of the origin of the forces for aster movements being dynein-dependent vesicle movements at the periphery of the asters. An attempt should be made to provide a more quantitative model to support the authors' claim more conclusively. For example, what does the model of the authors predict for the speed of aster movement as asters grow? Does one expect an increase of the speed of aster movement as a consequence of the aster surface increasing with time? Is the prediction of the model in agreement with the experimental data? How is this different from a model in which the force is generated throughout the aster? Does one expect different scaling for different types of models with different force generation mechanisms that could help to discriminate between models?

2) "We observed no outward movement of organelles when dynein was inhibited with CC1, so dynein is the dominant microtubule motor in egg asters." – but interestingly, in Video 13 one can see considerable back-and-forth movement of particles in the DIC channel, suggesting a tug-of-war of dynein and kinesins or myosins; when f-actin is fragmented, myosins cannot act as breaks, thus providing an explanation for faster organelle movement. It seems that the bead (artificial cargo) experiment supports the idea of tug-of-war, as beads only recruit dynein, and they move steadily toward the MTOC, while organelles may recruit different motors and after passing the aster surface, entering into a tug-of-war situation. Is it therefore justified to conclude that dynein is the dominant organelle motor? There should also be a better distinction, at least in the Discussion, between drag forces caused by network mesh density (e.g. f-actin) and effective viscosity due to transient interactions of dynein-opposing motors acting as breaks. The bead experiment seems to suggest that there is no difference in network-based viscosity between the aster surface and the interior, since the bead velocity is roughly constant.

3) The authors quantify aster movement in egg extract in clusters containing many asters, with obvious interaction between the different asters. This complicates the interpretation of the data since there are multiple external forces on each aster pair measured which have a strong influence that is ignored. Can the authors exclude that aster growth and aster pushing originating in the CPC-rich overlapping regions causes/contributes to aster movement. In this scenario dynein could simply contribute to holding all networks together that move together and other plus directed microtubule motors might be responsible for pushing asters apart. Vesicle movement at the aster periphery would then be a "consequence" and not the "cause" for aster movement in a region where the various networks might not be as crosslinked/as dense yet as in the more central regions of the aster. Laser cutting experiments could test this "pushing hypothesis". Laser cutting experiments (without and with dynein inhibition) could also test the existence of dynein-dependent stresses inside the aster that the authors predict to exist. This could support an important element of the authors' model. Analyzing the movements of asters with different numbers of neighboring asters in more or less "aster-dense" regions of the extract might be an alternative approach to test the alternative hypothesis of pushing forces playing a critical role in the movements of interconnected asters. Without such additional evidence, the current model of the authors remains rather speculative and some of the claims are not fully supported by the data.

4) How are the different boundary conditions in "closed" egg cells and "open" egg extract expected to affect the observed dynamics? Given the incompressibility of cytosol, how are aster movements and particularly cytosolic flows expected to differ between extract and egg cell? A more explicit discussion of how the results obtained from extract apply to the situation in a cell with a boundary will be useful.

---

## [Author Response]

Essential revisions:1) The authors need to provide a clearer biophysical picture supporting their model of the origin of the forces for aster movements being dynein-dependent vesicle movements at the periphery of the asters. An attempt should be made to provide a more quantitative model to support the authors' claim more conclusively. For example, what does the model of the authors predict for the speed of aster movement as asters grow? Does one expect an increase of the speed of aster movement as a consequence of the aster surface increasing with time? Is the prediction of the model in agreement with the experimental data? How is this different from a model in which the force is generated throughout the aster? Does one expect different scaling for different types of models with different force generation mechanisms that could help to discriminate between models?

In the previous submission we proposed that asters are positioned by surface forces and even put that conclusion in the title. We agree with the reviewers that our data do not prove this point. As summarized above, in the revised manuscript we emphasize flows, which we measured, and de-emphasize forces, which we only inferred. This major change in response to review was implemented throughout the paper, starting with the title. In the Discussion, we still discuss the hypothesis that pulling forces are exerted primarily on the aster periphery, but now using language that clearly represents it as one possible interpretation of the data. We are currently developing formal mechanical models for aster movement, in which forces and stresses on astral MTs cause the aster to move as well as deform, using hydrodynamic theories of active gels. These models require multiple figures to describe and are not yet fully developed. We feel they go beyond the scope of the current paper.

2) "We observed no outward movement of organelles when dynein was inhibited with CC1, so dynein is the dominant microtubule motor in egg asters." – but interestingly, in Video 13 one can see considerable back-and-forth movement of particles in the DIC channel, suggesting a tug-of-war of dynein and kinesins or myosins; when f-actin is fragmented, myosins cannot act as breaks, thus providing an explanation for faster organelle movement. It seems that the bead (artificial cargo) experiment supports the idea of tug-of-war, as beads only recruit dynein, and they move steadily toward the MTOC, while organelles may recruit different motors and after passing the aster surface, entering into a tug-of-war situation. Is it therefore justified to conclude that dynein is the dominant organelle motor? There should also be a better distinction, at least in the Discussion, between drag forces caused by network mesh density (e.g. f-actin) and effective viscosity due to transient interactions of dynein-opposing motors acting as breaks. The bead experiment seems to suggest that there is no difference in network-based viscosity between the aster surface and the interior, since the bead velocity is roughly constant.

This comment reflects multiple questions about ER transport in our system, including details of movement noted by reviewers on which we had not commented, and differences between what we report and previous papers that reported outward movement of ER by kinesin and tip tracking. To address these questions, we added new data (Figure 3—figure supplement 1), additional analysis of existing data (new panels in Figure 1—figure supplement 1), and new paragraphs in the Results and Discussion.

We suspected that our inability to detect outward movement of the ER in asters was due to our reliance on low magnification imaging (typically 20x objective), which tends to average movement over space and time. The new data and analysis report on image sequences collected with a 60x spinning disk confocal microscope, which reveals details of ER movement at short length and time scales that are averaged out in 20x videos. Revised Figure 1—figure supplement 1 and new Figure 3—figure supplement 1 reveal rapid, bidirectional movement and deformation of ER that may suggest tug-of-war between motors, and that we term “saltatory movement” in line with literature. To visualize saltatory movement of ER with respect to astral MTs, we developed a new radial kymograph analysis, performed in a rotating reference frame that drifts with the average ER flow estimated by PIV (Figure 1—figure supplement 1B). To show tangential co-movement of MTs, ER, and F-actin, we performed a new tangential kymograph analysis (Figure 1—figure supplement 1C). We also repeated PIV analysis for higher magnification videos, in which all networks tracked together as in 20x videos (Figure 3—figure supplement 1C). To relate examples of saltatory movement at smaller scales to net organelle transport at larger scales, we consider our mass transport analysis to be particularly important (Figure 7, Figure 7—figure supplement 1, Figure 7—figure supplement 2). Papers on organelle movement in egg extract focus on subsets of organelles that move, and they largely ignore stationary populations. Our new mass transport analysis quantitatively accounts for both moving and stationary organelles. Comparing between high and low spatiotemporal resolution, we were able to determine that rapid, saltatory motion was common but did not cause significant mass transport away from the MTOC. At larger scales, the ER either moves inwards or is stationary, even when dynein is inhibited.

Regarding dynein dominance in our system, our conclusion was based on predominantly inward transport at 20x and cessation of organelle transport when dynein was inhibited. We stand by our conclusion after adding the new high magnification analysis. Our observation that dynein dominates organelle transport in our system is consistent with previous reports, particularly a nice 1999 paper from Lane et al. In that study, Lane and Allen concluded dynein is the dominant ER motor in *Xenopus* egg extracts prepared from unfertilized eggs and that kinesin is inactive at least until the fifth interphase following fertilization. We added to the Discussion:

“We did not observe significant mass transport away from MTOCs by kinesins or tip tracking when dynein was inhibited, which argues against opposing motors. The predominance of dynein as the organelle motor in *Xenopus* eggs is consistent with previous studies using high magnification DIC imaging (Lane et al., 1999).”

Regarding possible changes in viscosity within the aster, we agree that the constant velocity of dynein-coated beads may suggest constant effective viscosity that is ~3.5x higher when F-actin is present. The role of MT and F-actin network turnover in permitting bead transport remains unclear. To interpret the cessation of organelle transport in the aster interior, we favor models where non-motor “brake” proteins engage organelles, rather than changes in viscosity. We modified the Discussion to more clearly reflect this hypothesis. Several candidate ER brake proteins are known, and we plan to test them in the future. That said, we cannot rule out the possibility that the more convoluted shape of the ER makes it more sensitive than beads to steric or hydrodynamic drag from astral MTs.

3) The authors quantify aster movement in egg extract in clusters containing many asters, with obvious interaction between the different asters. This complicates the interpretation of the data since there are multiple external forces on each aster pair measured which have a strong influence that is ignored. Can the authors exclude that aster growth and aster pushing originating in the CPC-rich overlapping regions causes/contributes to aster movement. In this scenario dynein could simply contribute to holding all networks together that move together and other plus directed microtubule motors might be responsible for pushing asters apart. Vesicle movement at the aster periphery would then be a "consequence" and not the "cause" for aster movement in a region where the various networks might not be as crosslinked/as dense yet as in the more central regions of the aster. Laser cutting experiments could test this "pushing hypothesis". Laser cutting experiments (without and with dynein inhibition) could also test the existence of dynein-dependent stresses inside the aster that the authors predict to exist. This could support an important element of the authors' model. Analyzing the movements of asters with different numbers of neighboring asters in more or less "aster-dense" regions of the extract might be an alternative approach to test the alternative hypothesis of pushing forces playing a critical role in the movements of interconnected asters. Without such additional evidence, the current model of the authors remains rather speculative and some of the claims are not fully supported by the data.

We thank and agree with the reviewers. Our evidence for pulling vs. pushing is based entirely on molecular inhibition in the case of asters moving apart on passivated surfaces, and on the location of dynein in the case of asters gliding on dynein-coated surfaces. In particular, simultaneously inhibiting dynein with CC1 and fragmenting F-actin with Cytochalasin D caused an almost complete block to aster movement in both systems. Laser cutting and magnetic tweezers are good suggestions but we lack the equipment to do this in house and also note that the manuscript is already long.

4) How are the different boundary conditions in "closed" egg cells and "open" egg extract expected to affect the observed dynamics? Given the incompressibility of cytosol, how are aster movements and particularly cytosolic flows expected to differ between extract and egg cell? A more explicit discussion of how the results obtained from extract apply to the situation in a cell with a boundary will be useful.

We thank and agree with the reviewers. In response, we have added new experimental data and analyses in new Figure 5—figure supplement 2. Even in egg extracts with open boundary conditions, vesicles visible in DIC exhibited saddle-shaped flow fields with respect to separating asters: the separating asters advected these vesicles perpendicular to the midzone and away from the midpoint between the asters, while vesicles along the midzone flowed toward the midpoint between the asters. This latter flow could be consistent with incompressibility of cytosol. We added a new paragraph to the Results:

“If separating asters on passivated coverslips also advect cytosol we would expect them to generate hydrodynamic forces and compensatory flows outside the asters. […] We indeed observed inward flow along the interaction zone (Figure 5—figure supplement 2).”, and we echoed in the Discussion where we describe Figure 9. Similar compensatory flows are described in the legend of Video 2 as well.

In eggs with closed boundary conditions, we hypothesize similar saddle-shaped flow fields, displaced around separating asters and into the midzone between them. In revising the cartoon model at the end of the manuscript we explicitly draw in these hypothetical flows with the goals of spurring new experiments in systems where the egg is transparent.

This reviewer comment highlights a related question, how do cytoplasmic networks move relative to one another after astral MTs fill the cell? This situation is especially relevant in smaller cells, but also relevant to later stages of aster separation in larger cells. While co-movement likely breaks down at least somewhere after astral MTs fill the cell, the view developed here of asters as a deformable and porous gel may still provide a helpful framework. When astral MTs fill the cell, the detailed viscous, elastic, and poroelastic properties of asters may all be relevant, and continued experiments and modeling are needed.